# Export flux of unprocessed atmospheric nitrate from temperate forested catchments: A possible new index for nitrogen saturation

Fumiko Nakagawa[1], Urumu Tsunogai[1], Yusuke Obata[1], Kenta Ando[1], Naoyuki Yamashita[2$], Tatsuyoshi Saito[2&], Shigeki Uchiyama[2#], Masayuki Morohashi[2], Hiroyuki Sase[2]

[1]Graduate School of Environmental Studies, Nagoya University, Furo-cho, Chikusa-ku, Nagoya 464-8601, Japan.
[2]Asia Center for Air Pollution Research, 1182 Sowa, Nishi-ku, Niigata-shi, Niigata 950-2144, Japan.
[$]Present address: Forestry and Forest Products Research Institute, Tsukuba, Ibaraki 305-8687, Japan.
[&]Present address: Niigata Prefectural Institute of Public Health and Environmental Sciences, 314-1 Sowa, Nishi-ku, Niigata-Shi, Niigata 950-2144, Japan.
[#]Present address: Ministry of Agriculture, Forestry and Fisheries, 1-2-1, Kasumigaseki, Chiyoda-ku, Tokyo 100-8950, Japan.

*Correspondence to*: Urumu Tsunogai (urumu@nagoya-u.jp)

**Abstract.** To clarify the biological processing of nitrate within temperate forested catchments using unprocessed atmospheric nitrate exported from each catchment as a tracer, we continuously monitored stream nitrate concentrations and stable isotopic compositions, including $^{17}$O-excess ($\Delta^{17}$O), in three forested catchments in Japan (KJ, IJ1, and IJ2) for more than two years. The catchments showed varying flux-weighted average nitrate concentrations: 58.4, 24.4, and 17.1 µmol L$^{-1}$ in KJ, IJ1, and IJ2, respectively, which correspond to varying export fluxes of nitrate: 76.4, 50.1, and 35.1 mmol m$^{-2}$ in KJ, IJ1, and IJ2, respectively. In addition to stream nitrate, nitrate concentrations and stable isotopic compositions in soil water were determined for comparison in the most nitrate-enriched catchment (the site KJ). While the $^{17}$O-excess of nitrate in soil water showed significant seasonal variation, ranging from +0.1 to +5.7‰ in KJ, stream nitrate showed small variation, from +0.8 to +2.0‰ in KJ, +0.7 to +2.8‰ in IJ1, and +0.4 to +2.2‰ in IJ2. We conclude

that the major source of stream nitrate in each forested catchment is groundwater nitrate. Additionally, the significant seasonal variation found in soil nitrate is buffered by the groundwater nitrate. The estimated annual export flux of unprocessed atmospheric nitrate accounted for $9.4\pm2.6\%$, $6.5\pm1.8\%$, and $2.6\pm0.6\%$ of the annual deposition flux of atmospheric nitrate in KJ, IJ1, and IJ2, respectively. The export flux of unprocessed atmospheric nitrate relative to the deposition flux showed a clear normal correlation with the flux-weighted average concentration of stream nitrate, indicating that reductions in the biological assimilation rates of nitrate in forested soils, rather than increased nitrification rates, are likely responsible for the elevated stream nitrate concentration, probably as a result of nitrogen saturation. The export flux of unprocessed atmospheric nitrate relative to the deposition flux in each forest ecosystem is applicable as an index for nitrogen saturation.

# 1 Introduction

## 1.1 Stream nitrate being exported from forested watersheds

Nitrate is one of the most important nitrogen nutrient for primary production in aquatic environments. As a result, an excess of nitrate in stream water can cause significant ecological and economic problems, such as eutrophication in downstream areas, including lakes, estuaries, and oceans (McIsaac et al., 2001; Paerl, 2009).

Forested ecosystems have traditionally been considered nitrogen-limited. However, because of elevated nitrogen loading through atmospheric deposition, some forested ecosystems become nitrogen-saturated (Aber et al., 1989), from which elevated levels of nitrate are exported (Peterjohn et al., 1996; Wright and Tietema, 1995). Either increased nitrification rates in forested soils or reductions in N retention are assumed to be responsible for both enhanced nitrogen leaching from soils and the increased export flux of nitrate in nitrogen-saturated watersheds (Peterjohn et al., 1996).

Nitrate concentrations in stream water are controlled through the complicated interplay between several processes within a catchment, including: (1) the addition of atmospheric nitrate ($NO_3^-{}_{atm}$) through deposition, (2) the production of nitrate through microbial nitrification in soils, (3) the removal of nitrate through assimilation by plants and microbes, and (4) the removal of nitrate
through dissimilatory nitrate reduction by microbes. Therefore, interpretation of the processes regulating nitrate concentrations in stream water is not always straightforward. The detailed processes to enhance nitrate concentrations in the streams eluted from nitrogen-saturated forested catchments have not yet been clarified.

The natural stable isotopic composition of nitrate ($\delta^{15}N$ and $\delta^{18}O$) has been widely used to
determine the origin and behaviour of nitrate in stream water (Durka et al., 1994; Kendall, 1998; Kendall et al., 2007). In addition to these traditional isotopes, $^{17}O$-excess ($\Delta^{17}O$; the definition will be presented in section 1.2) of nitrate has been used as an additional, more robust tracer for unprocessed $NO_3^-{}_{atm}$ (nitrate supplied via atmospheric deposition that has not been involved in the N cycle through the biological processing of nitrate, such as assimilation and denitrification, within
surface ecosystems) in stream water in recent years (Bourgeois et al., 2018b; Michalski et al., 2004; Riha et al., 2014; Sabo et al., 2016; Tsunogai et al., 2010; Tsunogai et al., 2014; Tsunogai et al., 2016). By determining the $^{17}O$-excess of stream nitrate, we can quantify the proportion of unprocessed $NO_3^-{}_{atm}$ within stream nitrate accurately and precisely. Additionally, by determining both the concentration and the $^{17}O$-excess of stream nitrate, we can quantify the concentration of
unprocessed $NO_3^-{}_{atm}$ in stream water (Tsunogai et al., 2014). Recent studies on unprocessed $NO_3^-{}_{atm}$ exported from forested catchments via streams during the base flow period have revealed that the export flux of unprocessed $NO_3^-{}_{atm}$ increases in accordance with increases in the stream nitrate concentration (Rose et al., 2015a; Rose et al., 2015b; Tsunogai et al., 2014). In addition, Tsunogai et al. (2014) successfully used the directly exported flux of unprocessed $NO_3^-{}_{atm}$ relative to the entire
deposition flux of $NO_3^-{}_{atm}$ as an index to evaluate the biological metabolic rate of nitrate in forest soils in catchment area. These results imply that unprocessed $NO_3^-{}_{atm}$ exported from forested

catchments can be used as a robust tracer to evaluate the biological processing of nitrate in each catchment area and to clarify the processes regulating nitrate concentrations in stream water.

In this study, we monitored both the concentrations and stable isotopic compositions (including $\Delta^{17}O$) of stream nitrate exported from three forested catchments in Japan for more than 2 years. The catchments showing various average nitrate concentrations in the streams were chosen for the targets in this study. In addition to nitrate in streams, the nitrate concentrations and stable isotopic compositions in soil water were determined over the same observation period for comparison in one catchment. Based on the differences in the direct export flux of unprocessed $NO_3^-{}_{atm}$ relative to the entire deposition flux of $NO_3^-{}_{atm}$ between the catchments, we aimed to clarify the processes regulating nitrate concentrations in stream water exported from temperate forested watersheds, with special emphasis on the relationship with nitrogen saturation. That is to say, through observation in this study, we will quantify the extent of changes in the biological metabolic processes of nitrate in temperate forested watersheds under nitrogen saturation, which show elevated export flux of nitrate.

## 1.2 $^{17}O$-excess of nitrate

The natural stable isotopic composition of nitrate is represented by its $\delta^{15}N$, $\delta^{17}O$, and $\delta^{18}O$ values. The delta ($\delta$) values are calculated as $R_{sample}/R_{standard} - 1$, where R is the $^{18}O/^{16}O$ ratio for $\delta^{18}O$ (or the $^{17}O/^{16}O$ ratio for $\delta^{17}O$ or the $^{15}N/^{14}N$ ratio for $\delta^{15}N$) in both the sample and the respective international standard (air $N_2$ for nitrogen and Vienna standard mean ocean water (VSMOW) for oxygen). Atmospheric nitrate ($NO_3^-{}_{atm}$), most of which is produced via photochemical reactions between atmospheric NO and $O_3$, can be characterised by the anomalous enrichment in $^{17}O$ compared to remineralized nitrate ($NO_3^-{}_{re}$), which is produced from organic nitrogen through general chemical reactions, including microbial N mineralization and microbial nitrification in the biosphere (Alexander et al., 2009; Michalski et al., 2003; Morin et al., 2008; Tsunogai et al., 2010; Tsunogai et al., 2016). Note that $NO_3^-{}_{re}$ also applies to atmospheric nitrate that has been involved in the N cycle, undergoing a full cycle of assimilation, remineralization, and nitrification. Using the $\Delta^{17}O$ signature (the magnitude of $^{17}O$-excess) defined by the following equation (Kaiser et al., 2007;

Miller, 2002), we can distinguish unprocessed $NO_3^-{}_{atm}$ ($\Delta^{17}O > 0‰$) from $NO_3^-{}_{re}$ ($\Delta^{17}O = 0‰$; Nakagawa et al., 2013):

$$\Delta^{17}O = \frac{1+\delta^{17}O}{\left(1+\delta^{18}O\right)^{\beta}} - 1, \tag{1}$$

where the constant $\beta$ is 0.5279 (Kaiser et al., 2007; Miller, 2002).

Continuous monitoring of the $\Delta^{17}O$ value of $NO_3^-{}_{atm}$ deposited at the mid-latitudes of East Asia has clarified that the annual average $\Delta^{17}O$ values of $NO_3^-{}_{atm}$ are almost constant at $26.6 \pm 0.9$ ‰ (the average and the $1\,\sigma$ variation range) (Nelson et al., 2018; Tsunogai et al., 2010; Tsunogai et al., 2016). In addition, $\Delta^{17}O$ is stable during the mass-dependent isotope fractionation processes within surface ecosystems (Miller, 2002; Thiemens et al., 2001). Therefore, while the $\delta^{15}N$ or $\delta^{18}O$

signature of $NO_3^-{}_{atm}$ can be overprinted by biological processes subsequent to deposition, $\Delta^{17}O$ can be used as a robust tracer of unprocessed $NO_3^-{}_{atm}$ to reflect its accurate mole fraction within total $NO_3^-$, regardless of partial metabolism (partial removal of nitrate through denitrification and assimilation) subsequent to deposition (Michalski et al., 2004; Tsunogai et al., 2011; Tsunogai et al., 2014), using the following equation:

$$\frac{C_{atm}}{C_{total}} = \frac{\Delta^{17}O}{\Delta^{17}O_{atm}}, \tag{2}$$

where $C_{atm}$ and $C_{total}$ denote the concentrations of unprocessed $NO_3^-{}_{atm}$ and $NO_3^-$ in each water sample, respectively, and $\Delta^{17}O_{atm}$ and $\Delta^{17}O$ denote the $\Delta^{17}O$ values of $NO_3^-{}_{atm}$ and total nitrate in each water sample, respectively. This is the primary advantage of using the $^{17}O/^{16}O$ ratio as an additional tracer of unprocessed $NO_3^-{}_{atm}$. In this study, we used the average $\Delta^{17}O$ value of $NO_3^-{}_{atm}$

obtained at the nearby Sado-seki monitoring station during the observation period from April 2009 to March 2012 ($\Delta^{17}O_{atm} = +26.3‰$; Tsunogai et al., 2016) for $\Delta^{17}O_{atm}$ in Eq. (2) to estimate $C_{atm}$ in the study streams, allowing an error range of 3‰, in which the factor changes in $\Delta^{17}O_{atm}$ from $+26.3‰$ caused by both areal and seasonal variation in the $\Delta^{17}O$ values of $NO_3^-{}_{atm}$ have been considered (Tsunogai et al., 2016).

Moreover, additional measurements of the $\Delta^{17}O$ values of nitrate together with $\delta^{18}O$ enable us to exclude the contribution of unprocessed $NO_3^-{}_{atm}$ in the determined $\delta^{18}O$ values and to estimate the corrected $\delta^{18}O$ values ($\delta^{18}O_{re}$) for accurate evaluation of the source and behaviour of $NO_3^-{}_{re}$, including anthropogenically produced $NO_3^-{}_{re}$ (Dejwakh et al., 2012; Liu et al., 2013; Riha et al.,

2014; Tsunogai et al., 2011; Tsunogai et al., 2010).

## 2 Experimental Section

### 2.1 Site description

In this study, we determined the export flux of unprocessed $NO_3^-{}_{atm}$ through monitoring of stream water in three forested catchments in Japan in which forest coverage rates exceed 99%: a catchment

(Site KJ) in the Kajikawa forested watershed and two subcatchments (Sites IJ1 and IJ2) in the Lake Ijira watershed (Fig. 1(a)). The deposition rate of $NO_3^-{}_{atm}$ was determined for each catchment by collecting samples of deposition outside the forest canopy. Soil water samples were also collected from the site KJ.

The site KJ is located in the northern part of Shibata city, Niigata Prefecture, near the coast of the

Japan Sea (Fig. 1(a)). The bedrock consists of granodiorite, and brown forest soils have developed (Kamisako et al., 2008; Sase et al., 2008). The forest is composed of Japanese cedars (*Cryptomeria japonica*), approximately 40 years old in 2012 (Sase et al., 2012). This site is characterised by perhumid climate conditions with no clear dry season during the year. The daily air temperature in the region varies from –2 °C to +34 °C, with an annual mean of 13 °C during the observation period

of this study. The annual mean precipitation is approximately 2500 mm, of which approximately 17% occurs during spring (from March to May), approximately 20% occurs during summer (from June to Aug.), approximately 28% occurs during fall (from Sep. to Nov.), and approximately 35% occurs during winter (from Dec. to Feb.). The site usually experiences snowfall from late December to March, with the maximum depth exceeding 100 cm, even on the slope. The studied catchment

area is 3.84 ha, with an elevation from 60 to 170 m above sea level (Fig. 1(b)). The catchment is characterised by a high loading rate of atmospheric nitrogen (more than 120 mmolN m$^{-2}$ yr$^{-1}$;

Kamisako et al., 2008), as well as elevated nitrate concentration (45 μmol $L^{-1}$ on average) in the stream water eluted from the catchment. In the same Niigata Prefecture in which KJ is located, Koshikawa et al. (2011) determined stream chemistry from streams (n=62) having various catchment areas, ranging from 0.7 ha to 1800 ha. They performed principal component analysis (PCA) of the various factors related to the stream chemistry including nitrate concentration, but could not find any significant relationship between stream nitrate concentration and catchment area. Thus, the differences in the catchment area (from 0.7 to 1,800 ha) had little impact on the stream nitrate concentration. Additionally, Kamisako et al. (2008) concluded that atmospheric nitrogen inputs are exceeding the biological demand at site KJ and proposed that the site was under nitrogen saturation (Aber et al., 1989). As a result, we chose this catchment to study unprocessed $NO_{3\ atm}^-$ as a tracer, as it is an example of a catchment enriched in stream nitrate, while the catchment area (3.94 ha) was relatively smaller than the other sites targeted in this study.

Lake Ijira (Fig. 2) is a reservoir constructed on one of the tributaries of the Nagara River in the Gifu prefecture, Honshu, Japan. The mean annual precipitation is approximately 3300 mm. The precipitation regime is characterised by relatively wet springs and summers (200 mm month$^{-1}$ from April to September) and relatively dry winters (approximately 100 mm month$^{-1}$ from December to February). The daily air temperature in the region varies from –3 °C to +31 °C, with an annual mean of 13 °C. The site is covered with snow from December to March every year.

The Kamagadani catchment (Site IJ1; 298 ha) and the Kobora catchment (Site IJ2; 108 ha) in the Lake Ijira watershed were studied (Fig. 2). The bedrock consists of chert (90%) and mudstone (10%) from the Middle Jurassic to Early Cretaceous age at these sites, and the dominant soil type is brown forest soils (Nakahara et al., 2010). The dominant vegetation in the Kamagadani catchment (IJ1) is Japanese cypress (*Chamaecyparis obtusa*, 49%), followed by broadleaf trees (29%), Japanese red pine (*Pinus densiflora*, 13%), and Japanese cedar (*Cryptomeria japonic*a, 8%), while the dominant vegetation in the Kobora catchment (IJ2) is Japanese red pine (*Pinus densiflora*, 46%), followed by broadleaf trees (30%), Japanese cypress (*Chamaecyparis obtusa*, 17%), and Japanese cedar (*Cryptomeria japonic*a, 7%). Japanese cypress and Japanese cedar stands are plantation

forests, with ages ranging from 15 to 25 years and 30 to 45 years, respectively, in 1998. The red pine and broadleaf stands are secondary forests. Major tree species in the secondary broadleaf forests are *Clethra barbinervis*, *Quercus serrata*, *Ilex pedunculosa*, *Quercus variabilis*, *Carpinus tschonoskii*, *Acer mono*, and *Quercus glauca*.

The annual wet deposition flux of $NO_3^-{}_{atm}$ in the Lake Ijira watershed was the highest of all EANET deposition monitoring sites in Japan (Yamada et al., 2007), probably because the catchment is located only approximately 40 km north of Nagoya and the surrounding industrial area (Chukyo industrial area). As a result, the discharge rate, water temperature, pH, electrical conductivity (EC), and alkalinity have been measured continuously at the outlets of the IJ1 and IJ2 (RW1 and RW3,

respectively, in Fig. 2) since 1988 (Nakahara et al., 2010). Nakahara et al. (2010) also proposed that Site IJ1 was under nitrogen saturation (stage 2; Aber et al., 1989) since 1997. For this reason, we chose the Lake Ijira watershed for this study. Details of the Lake Ijira watershed have been described in past studies (Nakahara et al., 2010; Yamada et al., 2007).

### 2.2 Stream water and discharge rates

Samples of stream water were collected manually in bottles that were rinsed at least twice with the sample itself at the outlet of each catchment (the weir in KJ, RW1 in IJ1, and RW3 in IJ2; Figs. 1 and 2) approximately once a month from May 2012 to December 2014 in KJ and from March 2012 to December 2014 in IJ1 and IJ2. In this study, 1-L or 2-L polyethylene bottles, washed using chemical detergents, rinsed at least thrice using deionised water, and then dried in the laboratory,

were used.

At the site KJ, a V-notch weir (half angle: 30°) and a partial flume were installed at the bottom of the catchment (Fig. 1(b)) where the stream water was collected. The data from the V-notch weir was used to measure the discharge rate. At the site IJ1, the discharge rates were calculated from both water depth and flow velocity at RW1 in Fig. 2. The water depth was measured at 100-cm intervals

across the river flow, and the flow velocity was measured at the midpoints of each 100-cm split using a flow meter (CM-10S, Toho Dentan, Tokyo, Japan). At the site IJ2, the discharge rates were

estimated from the calculated values from IJ1, assuming that the discharge rates from both sites varied in proportion to the area of the catchments.

## 2.3 Soil water

Soil water samples (n=45) were collected into 500-mL pre-evacuated glass bottles at two stations
(SLS and SMS; Fig. 1(b)) within the KJ catchment on average once every six weeks from December 2012 to December 2014, using porous cup soil solution samplers (DIK-8390-11/DIK-8390-58, DAIKI, Japan). Because the site is covered with snow in winter, however, a limited number of samples were taken between December and March (n = 9).

The SLS station is located by the stream side, while the SMS station is located approximately 20 m
away from the SLS station in the northeast direction (Fig. 1(b)). The SMS station is 23 m higher than the SLS station in altitude. The soil water samples were collected at a depth of 20 cm at each station (SLS 20 and SMS 20). Soil water samples were also collected at a depth of 60 cm at the SLS station (SLS 60).

## 2.4 Atmospheric nitrate deposition rates

For the site KJ, a filtering-type bulk deposition sampler with a funnel (200 mm diameter) installed in an open field outside the forest canopy on the northern ridge of the catchment (Fig. 1(b)) was used to determine the areal deposition flux of $NO_3^-{}_{atm}$ (Kamisako et al., 2008; Sase et al., 2008). Using the sampler, bulk depositions were collected into sample bottles at intervals of approximately four weeks. Sample bottles were covered with aluminium foil or enclosed in a polystyrene foam box
to avoid light and suppress algal growth during storage in the field. The volume of each sample was determined using plastic cylinders in the field, and portions of each sample were brought to the laboratory for further analysis. Please note that the dry deposition flux, especially for gaseous $HNO_3$, is underestimated in the $NO_3^-{}_{atm}$ deposition flux determined through this method (Aikawa et al., 2003), while the deposition flux of $NO_3^-{}_{atm}$ may be overestimated as a result of the progress of

nitrification in sample bottles during storage in the field until recovery (Clow et al., 2015). Errors in the deposition flux of $NO_3^-{}_{atm}$ will be discussed in section 3.1.

For the sites IJ1 and IJ2, data on the areal $NO_3^-{}_{atm}$ deposition flux, determined separately for wet and dry deposition at the outlet of IJ1 (140 m above sea level; Fig. 2) and reported by EANET
(EANET, 2014, 2015), were used in this study. The dry deposition flux was calculated from the concentrations of particulate nitrate and gaseous $HNO_3$ in air.

**2.5 Analysis**

Samples of stream water (KJ, IJ1, and IJ2), soil water (KJ), and deposition (KJ) were transported to the laboratory within 1 h after collection and were passed through a membrane filter (pore size, 0.45
μm) and stored in a refrigerator (4 °C) until chemical analysis was performed. The concentrations of $NO_3^-$ were measured by ion chromatography (DX-500, Dionex Inc., USA), together with major anions and cations. Samples were analyzed within a few weeks of sampling, then sealed in 50- or 100-mL polyethylene bottles for further analysis, including measurement of the isotopes in the stream and soil water samples reported in this study. Because the stream water samples were
analyzed for various components, the number of samples for measurement on the isotopes of $NO_3^-$ were approximately 1/2 of the entire stream water samples. Prior to isotope analysis, the $NO_3^-$ concentration of each stream water sample for measurement of the isotopes of $NO_3^-$ was determined again by ion chromatography to exclude samples that had been altered during storage. The longest storage period between bottling and isotope analysis was two years. None of the samples
determined in this study showed significant $NO_3^-$ deterioration or contamination during storage.

The $\delta^2H$ and $\delta^{18}O$ values of $H_2O$ in the stream and soil water samples were analyzed using the cavity ring-down spectroscopy method by employing an L2120-i instrument (Picarro Inc., Santa Clara, CA, USA) equipped with an A0211 vaporizer and autosampler. The errors (standard errors of the mean) in this method were ± 0.5‰ for $\delta^2H$ and ± 0.1‰ for $\delta^{18}O$. Both the VSMOW and
standard light Antarctic precipitation (SLAP) were used to calibrate the values to the international scale.

To determine the stable isotopic compositions of $NO_3^-$ in the stream and soil water samples, $NO_3^-$ in each sample was chemically converted to $N_2O$ using a method originally developed to determine the $^{15}N/^{14}N$ and $^{18}O/^{16}O$ ratios of seawater and freshwater $NO_3^-$ (McIlvin and Altabet, 2005) that was later modified (Konno et al., 2010; Tsunogai et al., 2008; Tsunogai et al., 2018; Yamazaki et al., 2011). In brief, 11 mL of each sample solution was pipetted into a vial with a septum cap. Then, 0.5 g of spongy cadmium was added, followed by 150 µL of a 1 M $NaHCO_3$ solution. The sample was then shaken for 18–24 h at a rate of 2 cycles/s. Then, the sample solution (10 mL) was decanted into a different vial with a septum cap. After purging the solution using high-purity helium, 0.4 mL of an azide/acetic acid buffer, which had also been purged using high-purity helium, was added. After 45 min, the solution was alkalinized by adding 0.2 mL of 6 M NaOH.

Then, the stable isotopic compositions ($\delta^{15}N$, $\delta^{18}O$, and $\Delta^{17}O$) of the $N_2O$ in each vial were determined using the continuous-flow isotope ratio mass spectrometry (CF-IRMS) system at Nagoya University. The analytical procedures performed using the CF-IRMS system were the same as those detailed in previous studies (Hirota et al., 2010; Komatsu et al., 2008). The obtained values of $\delta^{15}N$, $\delta^{18}O$, and $\Delta^{17}O$ for the $N_2O$ derived from the $NO_3^-$ in each sample were compared with those derived from our local laboratory $NO_3^-$ standards to calibrate the values of the sample $NO_3^-$ to an international scale and to correct for both isotope fractionation during the chemical conversion to $N_2O$ and the progress of oxygen isotope exchange between the $NO_3^-$-derived reaction intermediate and water (ca. 20%). The local laboratory $NO_3^-$ standards were calibrated using internationally distributed isotope reference materials (USGS-34 and USGS-35). In this study, we adopted the internal standard method (Nakagawa et al., 2013; Tsunogai et al., 2014; Tsunogai et al., 2018) for the calibration of sample $NO_3^-$.

To determine whether samples had deteriorated or were contaminated during storage and whether the conversion rate from $NO_3^-$ to $N_2O$ was sufficient, the concentration of $NO_3^-$ in the samples was determined each time we analyzed the isotopic composition using CF-IRMS, based on the $N_2O^+$ or $O_2^+$ outputs. We adopted the $\delta^{15}N$, $\delta^{18}O$, or $\Delta^{17}O$ values only when the concentration measured via CF-IRMS correlated with the concentration measured via ion chromatography prior to isotope

analysis within a difference of 10%. Approximately 10% of all isotope analyses showed conversion efficiencies lower than this criterion. The $NO_3^-$ in these samples was converted to $N_2O$ again and reanalyzed to determine stable isotopic composition.

We repeated the analysis of $\delta^{15}N$, $\delta^{18}O$, and $\Delta^{17}O$ values for each sample at least three times to
attain high precision. Most of the samples had a $NO_3^-$ concentration of greater than 10 μmol $L^{-1}$, which corresponded to a $NO_3^-$ quantity greater than 100 nmol in a 10-mL sample. This amount was sufficient for determining the $\delta^{15}N$, $\delta^{18}O$, and $\Delta^{17}O$ values with high precision. For cases where the $NO_3^-$ concentration was less than 10 μmol $L^{-1}$, the number of analyses was increased. Thus, all isotope values presented in this study have an error (standard error of the mean) better than ± 0.2‰
for $\delta^{15}N$, ± 0.3‰ for $\delta^{18}O$, and ± 0.1‰ for $\Delta^{17}O$.

Nitrite ($NO_2^-$) in the samples interferes with the final $N_2O$ produced from $NO_3^-$ because the chemical method also converts $NO_2^-$ to $N_2O$ (McIlvin and Altabet, 2005). Therefore, it is sometimes necessary to remove $NO_2^-$ prior to converting $NO_3^-$ to $N_2O$. However, in this study, all the stream and soil water samples analyzed for stable isotopic composition had $NO_2^-$ concentrations
lower than the detection limit (0.05 μmol $L^{-1}$). Because the minimum $NO_3^-$ concentration in the samples was 6.5 μmol $L^{-1}$ in this study, the $NO_2^-/NO_3^-$ ratios in the samples must be less than 0.8%. Thus, we skipped the processes for removing $NO_2^-$.

## 2.6 Possible variations in $\Delta^{17}O$ during partial removal and mixing

Because we used the power law shown in Eq. (1) for the definition of $\Delta^{17}O$, the $\Delta^{17}O$ values differ
from those based on the linear definition (Michalski et al., 2002). The differences in the $\Delta^{17}O$ values would have been less than 0.1‰ higher for the stream and soil water $NO_3^-$ if we had used the linear definition for calculation.

Compared with $\Delta^{17}O$ values based on the linear definition, $\Delta^{17}O$ values based on the power law definition are more stable during mass-dependent isotope fractionation processes, so we considered
the $\Delta^{17}O$ values of $NO_3^-$ to be stable, irrespective of any biological partial removal processes after deposition, such as assimilation or denitrification. Conversely, $\Delta^{17}O$ values based on the power law

definition are not conserved during mixing processes between fractions with different $\Delta^{17}O$ values, so the $C_{atm}/C_{total}$ ratio estimated using Eq. (2) deviates slightly from the actual $C_{atm}/C_{total}$ ratio in the samples. However, in this study, the extent of the deviations of the $C_{atm}/C_{total}$ ratios of the stream $NO_3^-$ was less than 0.2%, so we have disregarded this effect in the discussion.

**2.7 Calculation of the atmospheric nitrate export flux from each catchment**

To quantify the export flux of unprocessed $NO_3^-{}_{atm}$ from each catchment, the daily export flux of unprocessed $NO_3^-{}_{atm}$ per unit area of the catchment ($F_{atm}$) was calculated for each day on which the $\Delta^{17}O$ value of nitrate was determined, by applying equation (3) (Tsunogai et al., 2014):

$$F_{atm} = \frac{C_{atm} \times V}{S}, \tag{3}$$

where $C_{atm}$ denotes the concentration of unprocessed $NO_3^-{}_{atm}$, V denotes the daily average flow rate of stream water, and S denotes the total area of each catchment studied. The daily export fluxes of $NO_3^-$ ($F_{total}$) and $NO_3^-{}_{re}$ ($F_{re}$) per unit area of catchment were also calculated from the $NO_3^-$ concentration ($C_{total}$) and the daily average flow rate of the stream water (V) by applying equations (4) and (5):

$$F_{total} = \frac{C_{total} \times V}{S}, \tag{4}$$

$$F_{re} = F_{total} - F_{atm}, \tag{5}$$

Assuming $F_{atm}$ was stable during the period until the next observation ($\Delta t$), we can estimate the annual export flux of unprocessed $NO_3^-{}_{atm}$ per unit area of the catchment ($M_{atm}$) by integrating the $F_{atm}$ values for each year of observation using equation (6).

$$M_{atm} = \sum(F_{atm}(t) \times \Delta t), \tag{6}$$

We can also obtain the annual export flux for $NO_3^-$ ($M_{total}$) and $NO_3^-{}_{re}$ ($M_{re}$) by integrating $F_{total}$ and $F_{re}$ for each year of observation using equations (7) and (8).

$$M_{total} = \sum(F_{total}(t) \times \Delta t), \tag{7}$$

$$M_{re} = \sum(F_{re}(t) \times \Delta t), \tag{8}$$

By dividing $M_{atm}$ by the deposition flux of $NO_3^-{}_{atm}$ per unit area of the catchment, we can estimate the portion of $NO_3^-{}_{atm}$ deposited onto the catchment area that survived biological processing in the catchment basin.

$$\frac{M_{atm}}{D_{atm}} = \frac{\sum(F_{atm}(t) \times \Delta t)}{D_{atm}}, \tag{9}$$

where $D_{atm}$ denotes the annual deposition flux of $NO_3^-{}_{atm}$ per unit area of the catchment.

## 3 Results and Discussion

### 3.1 Site KJ: overview

The estimated annual discharge rate via the stream estimated by integrating the daily average flow rate of stream water (V) was 1,276 mm on average at site KJ during the observation undertaken
between 2013 and 2014. This value corresponds to 52% of the annual deposition rate determined at the meteorological station nearby (Nakajyo AMeDAS observatory; 2,454 mm on average between 2012 and 2014). Kamisako et al. (2008) determined the annual discharge rate at site KJ to be 1,439 mm during the observation undertaken between 2002 and 2007, using the same method as this study, and estimated that approximately 61% of the precipitation becomes stream outflow in this
catchment. Because the evapotranspiration loss from forested catchments in Japan was estimated to be 30% to 50% of deposition for the annual deposition rate from 2000 to 2500 mm (Ogawa, 2003), we concluded that the estimated annual discharge via the stream was highly reliable at the site, within the error range of 10%.

The determined export fluxes of nitrate in stream water ($F_{total}$) ranged from 74.7 to 698.4 µmol m$^{-2}$
day$^{-1}$, and the determined export fluxes of $NO_3^-{}_{atm}$ in stream water ($F_{atm}$) ranged from 3.3 to 46.1 µmol m$^{-2}$ day$^{-1}$ (Fig. 3(d)). We identified a clear increase in $F_{total}$ in winter, with the maximum flux occurring around December every year (Fig. 3(d)). A similar increase in the export fluxes of nitrate in winter was found in previous studies undertaken between 2002 and 2007 on the same stream (Kamisako et al., 2008). In accordance with the increase in $F_{total}$ in winter, $F_{atm}$ also increased.
Continuous monitoring of $\Delta^{17}O$ (Bourgeois et al., 2018a; Tsunogai et al., 2014) and $\delta^{18}O$ (Kendall

et al., 1995; Ohte et al., 2004; Pellerin et al., 2012; Piatek et al., 2005) of nitrate in past studies of streams eluted from forested catchments have often shown an increase in $F_{atm}$ during spring, probably because of $NO_3{}^-_{atm}$ accumulated in the snowpack discharging to the streams. At the site KJ, however, we could not find a significant $F_{atm}$ increase in spring.

The flux-weighted average stream nitrate concentration was 58.4 μmol $L^{-1}$. Compared with the average of 45.0 μmol $L^{-1}$ determined during past observations (Kamisako et al., 2008), a further increase in nitrate concentration was found at the site KJ in this study. Compared with the annual average stream nitrate concentrations eluted from forested catchments in Japan determined by Shibata et al. (2001) (n=18), that at site KJ corresponds to the highest, except for the two forested

catchments near metropolitan Tokyo showing high stream nitrate concentrations. The stable isotopic composition of stream nitrate differed from the concentration, showing only small temporal variation, from –3.2‰ to +1.6‰ for $\delta^{15}N$ (Fig. 3(b)), from –2.3‰ to +2.2‰ for $\delta^{18}O$ (Fig. 4), and from +0.8‰ to +2.0‰ for $\Delta^{17}O$ (Fig. 3(c)). The flux-weighted averages for the $\delta^{15}N$, $\delta^{18}O$ and $\Delta^{17}O$ values of nitrate were –2.2‰, +0.50‰, and +1.49‰, respectively. These values are typical for

nitrate exported from temperate forested watersheds (Bourgeois et al., 2018b; Nakagawa et al., 2013; Riha et al., 2014; Sabo et al., 2016; Tsunogai et al., 2014; Tsunogai et al., 2016).

Compared with the stream water, the soil water displayed higher nitrate concentrations, up to 1.6 mmol $L^{-1}$ (Fig. 5). The soil nitrate concentration showed significant seasonal variation irrespective of the locations or depths of sampling, with the maximum occurring in summer (August to

September) and minimum in winter (December) in our dataset (Fig. 5). Because we could not obtain data for soil water during January to March because of heavy snow at the site, nitrate concentration may be much lower during those months.

The stable oxygen isotopic compositions ($\delta^{18}O$ and $\Delta^{17}O$) of nitrate in the soil water also showed large seasonal variation, irrespective of the locations or depths of sampling, from –7.1‰ to +11.1‰

for $\delta^{18}O$ and from +0.1‰ to +5.7‰ for $\Delta^{17}O$ (Figs. 3(c) and 4), with the maximum occurring in winter and minimum in summer (Fig. 3(c)). In addition, the stable oxygen isotopic compositions ($\delta^{18}O$ and $\Delta^{17}O$) of nitrate showed a linear correlation on the $\Delta^{17}O$-$\delta^{18}O$ plot (Fig. 4). Because

$NO_3^-{}_{atm}$ is enriched in both $\Delta^{17}O$ and $\delta^{18}O$ and is the only possible source of nitrate with $\Delta^{17}O$ values higher than 0‰, mixing ratios between $NO_3^-{}_{atm}$ and $NO_3^-{}_{re}$ were primarily responsible for the variation in both $\Delta^{17}O$ and $\delta^{18}O$ in the soil nitrate (Costa et al., 2011). Moreover, the soil nitrate that was enriched during summer is mostly remineralized nitrate, produced through nitrification in

soils. The stable nitrogen isotopic composition ($\delta^{15}N$) of nitrate in the soil water samples also showed a larger temporal variation compared to the stream water nitrate, from −8.2‰ to +0.5‰ (Fig. 3(b)).

The areal bulk deposition flux of $NO_3^-{}_{atm}$ determined for the site KJ was 0.125 mmol m$^{-2}$ day$^{-1}$ (45.6 mmol m$^{-2}$ yr$^{-1}$ = 6.4 kgN ha$^{-1}$ yr$^{-1}$) on average during the observation period. As presented in

section 2.4, the deposition flux could be either underestimated, because of insufficient inclusion of the dry deposition flux (Aikawa et al., 2003) or overestimated, because of the progress of nitrification in sample bottles during storage in the field until recovery (Clow et al., 2015). Nevertheless, the deposition flux almost corresponds to the average areal total (wet + dry) deposition flux of atmospheric nitrate determined at the nearby Sado-seki National Acid Rain

Monitoring Station on Sado Island (38°14'59"N, 138°24'00"E; Fig. 1(a)) in 2013 (49.2 mmol m$^{-2}$ yr$^{-1}$ = 6.9 kgN ha$^{-1}$ yr$^{-1}$) and 2014 (48.3 mmol m$^{-2}$ yr$^{-1}$ = 6.8 kgN ha$^{-1}$ yr$^{-1}$), in which the wet deposition flux of nitrate (30.6 and 27.1 mmol m$^{-2}$ yr$^{-1}$ in 2013 and 2014, respectively), dry deposition flux of gaseous HNO$_3$ (13.5 and 15.3 mmol m$^{-2}$ yr$^{-1}$ in 2013 and 2014, respectively), and dry deposition flux of particulate nitrate (5.1 and 5.9 mmol m$^{-2}$ yr$^{-1}$ in 2013 and 2014, respectively)

were integrated (EANET, 2014, 2015). As a result, we use the bulk deposition flux determined in this study (45.6 mmol m$^{-2}$ yr$^{-1}$) as the areal total (wet + dry) deposition flux of $NO_3^-{}_{atm}$ ($D_{atm}$) at the site KJ by allowing an error range of 10%.

### 3.2 Sites IJ1 and IJ2: overview

The estimated annual discharge rate via the streams estimated by integrating the daily average flow
rates of stream water (V) was 2,057 mm on average at the sites during the observation. This value corresponds to 62% of the annual deposition rate (3,310 mm on average during the observation

undertaken between 2013 and 2014). Because the evapotranspiration loss from forested catchments in Japan was estimated to be 30% to 40% of deposition for the annual deposition rate of 3000 mm (Ogawa, 2003), we concluded that the estimated annual discharge via the stream was highly reliable in the sites, within the error range of 10%.

The determined export fluxes of nitrate in stream water ($F_{total}$) ranged from 39.3 to 293 µmol m$^{-2}$ day$^{-1}$ and from 26.1 to 267 µmol m$^{-2}$ day$^{-1}$ in IJ1 and IJ2, respectively, and the determined export fluxes of $NO_3^-{}_{atm}$ in stream water ($F_{atm}$) ranged from 1.6 to 18.3 µmol m$^{-2}$ day$^{-1}$ and from 0.75 to 12.3 µmol m$^{-2}$ day$^{-1}$ in IJ1 and IJ2, respectively (Fig. 6(d)). The values ranged from 13.6 to 58.4 µmol L$^{-1}$ for nitrate concentration (Fig. 6(a)), –2.2‰ to +5.0‰ for $\delta^{15}N$ (Fig. 6(b)), +1.0‰ to
+9.8‰ for $\delta^{18}O$, and +0.7‰ to +2.8‰ for $\Delta^{17}O$ (Fig. 6(c)) in IJ1, and from 11.1 to 60.9 µmol l L$^{-1}$ for nitrate concentration (Fig. 6(a)), –1.1‰ to +3.3‰ for $\delta^{15}N$ (Fig. 6(b)), -2.1‰ to +8.0‰ for $\delta^{18}O$, and +0.4‰ to +2.2‰ for $\Delta^{17}O$ (Fig. 6(c)) in IJ2.

Different from the site KJ, we could not find any clear seasonal variation in the concentration of nitrate, the stable isotopic compositions of nitrate, or the export fluxes of nitrate ($F_{total}$) and $NO_3^-{}_{atm}$
($F_{atm}$) in the stream water from IJ1 and IJ2. We could not identify a spring maximum in these catchments either. Conversely, we did find sporadic, short-term increases in nitrate of approximately 40 µmol L$^{-1}$ during the observation period. The increases were observed simultaneously at IJ1 and IJ2. Similar sporadic increases in nitrate concentration were found in Aug. 1994 during observations from 1988 to 2003 on the stream IJ1 (Nakahara et al., 2010). Except for
the sporadic, short-term increases in nitrate concentration, the stream water nitrate concentration and isotopic composition were almost constant at each site during the observation period. The flux-weighted average for the $\delta^{15}N$, $\delta^{18}O$ and $\Delta^{17}O$ values of stream nitrate were +0.23‰, +3.76‰, and +1.50‰ at IJ1, respectively, and +0.42‰, +1.57‰, and +0.85‰ at IJ2, respectively. These values are typical for nitrate exported from temperate forested watersheds (Bourgeois et al., 2018b;
Nakagawa et al., 2013; Riha et al., 2014; Sabo et al., 2016; Tsunogai et al., 2014; Tsunogai et al., 2016).

One of the striking features of the stream nitrate concentration at these sites is that nitrate concentrations at IJ1 were approximately $7\pm5$ µmol $L^{-1}$ higher than those for IJ2 determined at the same time throughout the observation period. Amongst the 71 pairs of data points, the reverse relationship (lower nitrate concentration in IJ1 compared with IJ2) was found only three times (Aug. 2012, July 2013, and Sep. 2013). Even during the sporadic, short-term increases in nitrate, the nitrate concentrations in IJ1 were generally higher than IJ2. Furthermore, not only the stream nitrate concentration but also the $\Delta^{17}O$ values of nitrate at IJ1 were higher than those at IJ2 (Fig. 6(c)). Amongst the 38 pairs of data points, the reverse relationship (lower $\Delta^{17}O$ values of nitrate in IJ1 compared with IJ2) was found only five times.

The flux-weighted average stream nitrate concentrations during the observation period were 24.4 and 17.1 µmol $L^{-1}$ in IJ1 and IJ2, respectively. Compared with the annual average stream nitrate concentrations eluted from forested catchments in Japan that were determined by Shibata et al. (2001) (n=18), those at sites IJ1 and IJ2 correspond to the 8th and 9th highest concentration, respectively. While the stream nitrate concentration in IJ1 showed an increasing trend year to year,
from 22 µmol $L^{-1}$ in the late 1980s to 42 µmol $L^{-1}$ in the early 2000s (Nakahara et al., 2010), the recent result (almost stable at 24.4 µmol $L^{-1}$ on average during the observation undertaken between 2012 and 2014; Fig. 6(a)) revealed that the trend in stream nitrate concentration had already changed from increasing to decreasing.

The areal deposition flux of $NO_3{}^-{}_{atm}$ was 0.122 mmol $m^{-2}$ $day^{-1}$ (44.5 mmol $m^{-2}$ $yr^{-1}$) on average
during the observation period (EANET, 2014, 2015). This value almost corresponds with the observed value from the KJ monitoring site.

### 3.3 Origin of stream nitrate in Site KJ

The runoff paths of water from the forested slope to the stream can be classified into (1) overland flow, (2) through flow (shallow subsurface flow above the water table), and (3) groundwater flow
(movement through the saturated zone) (Berner and Berner, 1987). The $\Delta^{17}O$ values of stream nitrate (+0.8 to 2.0‰) indicated that the major portion of stream nitrate was remineralized nitrate

($NO_{3\ re}^-$), produced through nitrification in soils, and thus unprocessed atmospheric nitrate ($NO_{3\ atm}^-$) contributed a minor portion of the total nitrate. This means that nitrate supplied via overland flow was a minor portion of stream nitrate. While stream nitrate showed similar $\Delta^{17}O$ values to soil nitrate (nitrate in the soil water samples of SLS20, SLS60, and SMS20), the variation in stream

nitrate was much smaller than soil nitrate (Fig. 4); from +0.8 to +2.0‰ for stream nitrate, while from +0.1‰ to +5.7‰ for soil nitrate. Because $\Delta^{17}O$ is stable during partial metabolism in soils (such as assimilation and denitrification), the present results imply that nitrate in the catchment groundwater was the major source of stream nitrate, while nitrate in through flow, in which the $\Delta^{17}O$ values must be similar with those of soil nitrate, was a minor contributor to the stream nitrate.

That is, while the $\Delta^{17}O$ values of soil nitrate represented the original $\Delta^{17}O$ values of nitrate now in the groundwater and the stream water, the large seasonal variation in the $\Delta^{17}O$ values of soil nitrate was buffered by nitrate reserves in the groundwater (Kabeya et al., 2007; Tsunogai et al., 2016). Therefore, little seasonal variation in the $\Delta^{17}O$ values of stream nitrate and only a small increase in $F_{atm}$ during spring were observed.

This hypothesis was supported by the $\delta^2H$, $\delta^{18}O$, and d-excess (=$\delta^2H - 8\times\delta^{18}O$; Dansgaard, 1964) values of stream and soil water. The values of $\delta^2H$, $\delta^{18}O$, and d-excess in stream water showed little temporal variation; –48.6±3.0 ‰, –9.1±0.3 ‰, and +24.2±1.9 ‰, respectively (the average and the 1 σ variation range of each), while larger temporal variation was seen in the corresponding values in soil water (Fig. S1). The values of $\delta^2H$, $\delta^{18}O$, and d-excess in rain (and snow) water in these regions

(Japan sea side of eastern Japan) shows large seasonal variation every year. In the case of d-excess, for instance, d-excess values of greater than +30‰ in winter and less than +10‰ in summer are seen in the rain water in these region (Tanoue et al., 2013). As a result, the observed large temporal variation in soil water reflected the large temporal variation in rain (and snow) water. Conversely, the small seasonal variation found in the values of $\delta^2H$, $\delta^{18}O$, and d-excess in stream water indicates

that the large temporal variation in rain (and snow) and soil water was buffered by groundwater. Additionally, the contribution of both overland flow and through flow should be minor in the stream.

This hypothesis was supported by the $\delta^{18}O$ values of nitrate as well. While the $\delta^{18}O$ values of nitrate could change during partial metabolism, the range of $\delta^{18}O$ variation in stream nitrate (-2.3 to +2.2‰) was within the range of soil nitrate (-7.1 to +11.1‰) (Fig. 4). In addition, stream nitrate data were plotted along the hypothetical mixing line between $NO_3^-{}_{atm}$ and $NO_3^-{}_{re}$ for soil nitrate (Fig. 4). We concluded that soil nitrate was the primary source of stream nitrate, but the temporal variation in the concentration and isotopic compositions of soil nitrate had been buffered by the huge nitrate reserve in the groundwater.

By extrapolating the linear correlation between $\Delta^{17}O$ and $\delta^{18}O$ in stream and soil nitrate shown in Fig. 4 ($r^2 = 0.647$, $p < 0.001$) to $\Delta^{17}O = 0$‰, we obtained the $\delta^{18}O$ value of $-2.7\pm0.6$ ‰ as the average $\delta^{18}O$ value of $NO_3^-{}_{re}$ in both stream and soil water. The $\delta^{18}O$ value of $NO_3^-{}_{re}$ correlated strongly with that of $NO_3^-{}_{re}$ being exported from forested catchments, for example, $NO_3^-{}_{re}$ exported from cool-temperate forested watersheds in Rishiri Island ($\delta^{18}O = -4.2\pm2.4$‰), where the $\delta^{18}O(H_2O)$ was approximately $-13$‰ (Tsunogai et al., 2010), $NO_3^-{}_{re}$ exported from a cool-temperate forested catchment in Teshio ($\delta^{18}O = -3.6\pm0.7$‰), where the $\delta^{18}O(H_2O)$ was approximately $-11$‰ (Tsunogai et al., 2014), and $NO_3^-{}_{re}$ exported from the temperate forested watersheds around Lake Biwa ($\delta^{18}O = -2.9\pm1.2$‰), where the $\delta^{18}O(H_2O)$ was $-7.8\pm1.0$‰ (Tsunogai et al., 2016).

The possible $\delta^{18}O$ value of $NO_3^-{}_{re}$ produced through microbial nitrification can be estimated using the equation shown below (Buchwald et al., 2012):

$$\delta^{18}O\left(NO_3^-{}_{re}\right) = \left\{\frac{2}{3} + \frac{1}{3}x\right\}\delta^{18}O_{H_2O} + \frac{1}{3}\left\{\left\{\delta^{18}O_{O_2} - 20.4\times10^{-3}\right\}\times(1-x) - 8.6\times10^{-3}\right\} +$$
$$\frac{2}{3}\times12.5\times10^{-3}\times x, \tag{10}$$

where $\delta^{18}O_{H_2O}$ denotes the $\delta^{18}O$ value of $H_2O$ during nitrification, $\delta^{18}O_{O_2}$ denotes the $\delta^{18}O$ value of $O_2$ during nitrification (+24.2‰ in this study), and $x$ denotes the amount of O atom exchange between nitrite and $H_2O$ during nitrification. By changing $x$ from 0 (no exchange) to 1 (full exchange), we can estimate the possible $\delta^{18}O$ value of $NO_3^-{}_{re}$ produced through microbial nitrification under an $H_2O$ of $-9.1$‰ (the average $\delta^{18}O$ value of $H_2O$ in the stream water samples; Fig. S1) as $-5.7\pm2.0$‰. Because the partial metabolism of nitrate would enhance the $\delta^{18}O$ values of

residual nitrate to some extent, the possible lowermost $\delta^{18}O$ value of $NO_3^-{}_{re}$ ($-7.7‰$) is the most probable $\delta^{18}O$ value of $NO_3^-{}_{re}$ originally produced through microbial nitrification in the forested soils at site KJ to explain the linear relation between $\Delta^{17}O$ and $\delta^{18}O$ values of both soil and stream nitrate shown in Fig. 4. Additionally, the observed average $\delta^{18}O$ value ($-2.7\pm0.6$ ‰), showing a
small difference from the possible lowermost original $\delta^{18}O$ value of $NO_3^-{}_{re}$ in both the stream and soil water, implies that $^{18}O$-enrichment through partial metabolism subsequent to the production of $NO_3^-{}_{re}$ was small, only $+5‰$ or less on the average in the forested soils in KJ. The relationship between $\Delta^{17}O$ and $\delta^{18}O$ of nitrate shown in Fig. 4 is highly useful for determining the $\delta^{18}O$ value of $NO_3^-{}_{re}$ in each catchment and thus the behaviour of produced $NO_3^-{}_{re}$ within the catchment
(Tsunogai et al., 2010).

**3.4 Seasonal variation at the site KJ**

Nitrate at the site KJ presented a clear export flux ($F_{total}$) increase in winter (Fig. 3(d)). High precipitation in winter is partially responsible for the increase in the export flux of water and thus the $F_{total}$ increase in winter. However, it is difficult to explain a nitrate concentration of greater than
80 µmol $L^{-1}$ only by higher precipitation in winter. Kamisako et al. (2008) found the same trend during their observation period from 2002 to 2007 at the same site and proposed that active biological assimilation of nitrate during the growing season was responsible for the nitrate concentration decrease in summer, and thus the nitrate concentration increase in winter. However, the present study revealed that the soil nitrate showed the opposite trend: a nitrate concentration
increase in summer and nitrate concentration decrease in winter, probably because of active nitrification in the soil in summer (Breuer et al., 2002; Hoyle et al., 2006; Tsunogai et al., 2014; Zaman and Chang, 2004). A clear decrease in the $\Delta^{17}O$ values of soil nitrate in summer (Fig. 3(c)) also supports the occurrence of active nitrification in summer (Tsunogai et al., 2014) because the $\Delta^{17}O$ values of remineralized nitrate produced through nitrification are $0‰$ (Michalski et al., 2004;
Nakagawa et al., 2013). Moreover, if such biological assimilation was responsible for the decrease in nitrate concentration in summer, enrichment in the values of $\delta^{15}N$ and $\delta^{18}O$ could be expected in

the residual portion of nitrate exported into the stream, while we could not find significant enrichment in summer (Figs. 3 and 4). As a result, it is difficult to assume active biological assimilation of nitrate in summer as responsible for the seasonal variation in stream nitrate concentration.

As presented in section 3.3, the major source of stream nitrate is likely groundwater nitrate that has been recharged by soil nitrate. The residence time of groundwater was estimated to be a few months for most of the catchments in Japan with a humid temperate climate using the deuterium excess as a tracer (Kabeya et al., 2007; Takimoto et al., 1994). While the soil nitrate concentration showed an increase in summer and decrease in winter, stream nitrate samples taken at the same time showed

the opposite trend (Fig. 7). However, if we assume a time lag of four months between the samples, as presented in Fig. 7, stream nitrate concentration shows a normal correlation with soil nitrate concentration ($r^2 = 0.41$ and $p < 0.03$ for SLS20, $r^2 = 0.37$ and $p < 0.03$ for SMS20).
The small increase/decrease in the $\Delta^{17}O$ values of stream nitrate can be explained by the increase/decrease in the $\Delta^{17}O$ values of soil nitrate four months earlier. This delay time reflects the

magnitude and flow of the nitrate reservoir in the groundwater of this catchment. We conclude that active nitrification in summer is largely responsible for the increase in stream nitrate concentration in winter, by increasing the nitrate concentration in groundwater that reflects nitrate accumulation over a few months prior to the observation.

### 3.5 The export flux of atmospheric nitrate and the relationship with nitrogen saturation

As already implied in previous studies at the site KJ (Kamisako et al., 2008; Sase et al., 2015), stream nitrate at the site KJ is characterised by elevated nitrate concentrations. Additionally, the stream water at the site IJ1 is characterised by nitrate concentrations higher than the stream water at the site IJ2.  The flux-weighted annual average stream nitrate concentration determined in this study was 58.4 $\mu mol\ L^{-1}$ at the site KJ, and 24.4 and 17.1 $\mu mol\ L^{-1}$ at the sites IJ1 and IJ2, respectively

(Table 1). The annual export flux of nitrate per unit area of the catchment ($M_{total}$) from the site KJ (76.4 mmol $m^{-2}\ yr^{-1}$) was also higher than the fluxes from the sites IJ1 and IJ2 (50.1 and 35.1 mmol

m$^{-2}$ yr$^{-1}$, respectively). In accordance with the variation in the export flux of nitrate, the unprocessed $NO_3^-{}_{atm}$ per unit area of the catchment ($M_{atm}$) also varied: 4.26±0.78 (mmol m$^{-2}$ yr$^{-1}$) from KJ, 2.88±0.52 (mmol m$^{-2}$ yr$^{-1}$) from IJ1, and 1.15±0.13 (mmol m$^{-2}$) from IJ2 (Table 1). As a result, not only the export flux of $NO_3^-{}_{re}$ produced through nitrification in forested soils but also the direct

drainage flux of unprocessed $NO_3^-{}_{atm}$ increased in accordance with the increases in the export flux of nitrate between the catchments.

Because the differences in the deposition flux of $NO_3^-{}_{atm}$ ($D_{atm}$) were small between the studied catchments (Table 1), regional changes in $D_{atm}$ cannot be the direct cause of the observed variation in $M_{atm}$ in accordance with variation in the stream nitrate concentrations. Moreover, the $M_{atm}/D_{atm}$

ratios estimated using the equation (9) also varied in accordance with the stream nitrate concentrations (Fig. 8(a)): 9.4±2.6% at the site KJ, 6.5±1.8% at the site IJ1, and 2.6±0.6% at the site IJ2, and thus the residual portion (90.6±2.6% in KJ, 93.5±1.8% in IJ1, and 97.4±0.6% in IJ2) underwent biological processing (such as assimilation and denitrification) before being exported from the surface ecosystem. The $M_{atm}/D_{atm}$ ratio, the directly exported flux of unprocessed $NO_3^-{}_{atm}$

relative to the entire deposition flux of $NO_3^-{}_{atm}$ in a catchment area, was used in our previous study as an index to evaluate the biological metabolic rate of nitrate in forested soils (Tsunogai et al., 2014), because the $(D_{atm} - M_{atm})/D_{atm}$ ratio (almost equal to the biological assimilation rate of $NO_3^-{}_{atm}$ relative to deposition rate of $NO_3^-{}_{atm}$ in a catchment; Tsunogai et al., 2014) increases in accordance with the decrease in biological metabolic rate of nitrate in forested soils (Fig. 9). The

normal correlation between stream nitrate concentrations and the $M_{atm}/D_{atm}$ ratios is an important finding to interpret the changes in stream nitrate concentrations between the catchments.

Rose et al. (2015a) determined $M_{atm}$ in forested catchments under various nitrogen saturation stages and found similar $M_{atm}$ variation in accordance with stream nitrate concentrations. When we estimated $M_{atm}/D_{atm}$ ratios for the catchments studied in Rose et al. (2015a) and plotted them as a

function of the stream nitrate concentration in Fig. 8(a) together with our data, both results plotted on the same region, showing a clear increasing trend in the $M_{atm}/D_{atm}$ ratios in accordance with

increases in the stream nitrate concentration and thus increases in the stage of nitrogen saturation (Fig. 8(a)).

Either increased nitrification rates in forested soils or reductions in the N retention ability are assumed to be responsible for enhanced nitrogen leaching from soils and the increased export flux

of nitrate in nitrogen-saturated catchments (Peterjohn et al., 1996). In the studied catchments, however, it is not possible to explain the variation in the export flux of unprocessed $NO_{3\ atm}^{-}$ between the catchments only by the variation in the nitrification rates in forested soils because the $M_{atm}/D_{atm}$ ratios are stable during the progress of nitrification in forested soils (Fig. 9). In Fig. 9, all the arrows (=flows) related to the determination on the $M_{atm}/D_{atm}$ ratios are shown in red/pink, while

the arrows (=flows) related to nitrification in soils are shown in brown/yellow. As represented by the differences in the colours, the $M_{atm}/D_{atm}$ ratios were determined independent of nitrification. Rather, varying N retention abilities (varying biological assimilation rates of nitrate, especially) in forested soils are required to explain the observed variation in the stream nitrate concentration and $M_{atm}/D_{atm}$ ratios between the catchments simultaneously (Fig. 9).

The present results imply that the major impact of nitrogen saturation was on the biological assimilation processes of nitrate, rather than the biological nitrification processes in soils. Furthermore, in addition to the stream nitrate concentration, the $M_{atm}/D_{atm}$ ratio in each forested catchment can be used as an index for the nitrogen saturation stage. That is, the studied catchments were under nitrogen saturation in the stage order of KJ > IJ1 > IJ2 (Fig. 8(a)).

Kamisako et al. (2008) reported that the deposition rate of atmospheric nitrogen in site KJ was one of the highest levels in forested catchments in Japan and exceeds the threshold for nitrogen saturation proposed by previous studies in Europe and the U.S. (Aber et al., 2003; Wright and Tietema, 1995). Kamisako et al. (2008) also found acidification of stream water during the periods with high concentrations of stream $NO_3^{-}$ and proposed that site KJ was under nitrogen saturation as

a result of the elevated deposition rate of atmospheric nitrogen. Nakahara et al. (2010) also proposed that site IJ1 has been under nitrogen saturation (stage 2) since 1997, based on observation of the atmospheric deposition rates, soil chemistry, stream water chemistry, and forest growth determined

at the site. Our conclusion based on the $M_{atm}/D_{atm}$ ratios is supported by these past studies performed at the sites.

All nitrate other than unprocessed $NO_3^-{}_{atm}$ can be classified as $NO_3^-{}_{re}$, including nitrate produced through natural or anthropogenic processes in the biosphere, hydrosphere, and geosphere, and
nitrate stored in soil, fertiliser, manure, and sewage. Therefore, except for those accompanied by secondary changes in biological assimilation processes of nitrate in forested soils, an increase in stream nitrate concentration resulting from artificial nitrate contamination processes in forested catchments does not increase $M_{atm}/D_{atm}$ ratios. As a result, the $M_{atm}/D_{atm}$ ratio in each forested catchment can be used as an index to differentiate increase in stream nitrate concentration because
of changes in biological assimilation processes of nitrate, from an increase in stream nitrate concentration resulting from nitrate contamination processes.

Stoddard (1994)  proposed the disappearance of seasonality in stream nitrate concentrations as an index for nitrogen saturation in forest ecosystems. However, because the seasonal changes in forested soils are buffered by groundwater in humid temperate climates such as Japan, the
seasonality in stream nitrate concentrations is not clear even when exported from "normal" forest (i.e., forest under stage zero of nitrogen saturation) (Mitchell et al., 1997). As a result, seasonality is not a reliable index of nitrogen saturation in forests in humid temperate climates. The present study implies that the $M_{atm}/D_{atm}$ ratio in each forested catchment, estimated from the [17]O-excess of stream nitrate, can be a robust, alternative index for the stage of nitrogen saturation irrespective of the
humidity of the climate.

To estimate $M_{atm}/D_{atm}$ ratios in a catchment, the export flux of nitrate ($M_{total}$), the [17]O-excess of stream nitrate, and the deposition rate of $NO_3^-{}_{atm}$ ($D_{atm}$) must be estimated. The deposition rate of $NO_3^-{}_{atm}$ ($D_{atm}$), however, is a difficult parameter to determine in forested catchments in general. An alternative parameter that we can determine more easily is the average concentration of $NO_3^-{}_{atm}$ in
stream water ($[C_{atm}]_{avg}$); therefore, we plotted $[C_{atm}]_{avg}$ as a function of the average concentration of nitrate ($[C_{total}]_{avg}$) in Fig. 8(b). While the correlation coefficient was poorer than the $M_{atm}/D_{atm}$ ratio, $[C_{atm}]_{avg}$ also presented a normal correlation with the concentration of stream nitrate (Fig. 8(b)),

probably because the differences in (1) $NO_3^-{}_{atm}$ concentration in wet deposition, (2) the dry deposition flux of $NO_3^-{}_{atm}$, and (3) the evaporative loss flux of water deposited onto forested soils were small within the catchments. As a result, in forested catchments where we can assume the differences in (1), (2), and (3) from the studied catchments are minimal, we can use $[C_{atm}]_{avg}$ as an

alternative but less reliable index of the stage of nitrogen saturation, instead of the $M_{atm}/D_{atm}$ ratios. Previous studies also found that the relative mixing ratios of unprocessed $NO_3^-{}_{atm}$ to total nitrate ($M_{atm}/M_{total}$ ratios) increased in proportion to the extent of both forest decline (Durka et al., 1994) and strip-cutting (Tsunogai et al., 2014). In the present study, however, we could not find clear changes in the $M_{atm}/M_{total}$ ratios between the catchments: 5.6% for the site KJ, 5.7% for the site IJ1,

and 3.3% for the site IJ2 (Table 1). Rose et al. (2015a) also reported that the $M_{atm}/M_{total}$ ratios were almost the same between forested catchments, irrespective of changes in their nitrogen saturation stages. While the annual export flux of nitrate and $NO_3^-{}_{atm}$ per unit area of the catchment increased by 6 and 20 times, respectively, in accordance with strip-cutting (Tsunogai et al., 2014), increases in $M_{total}$ and $M_{atm}$ in KJ compared with IJ2 were only 3 and 4 times, respectively, so we could not find

clear changes in $M_{atm}/M_{total}$ ratios between the catchments. Even in forested catchments where it is difficult to determine $D_{atm}$, the $M_{atm}/M_{total}$ ratio is not a suitable alternative index to the $M_{atm}/D_{atm}$ ratio for the stages of nitrogen saturation.

Rose et al. (2015a) also found a linear correlation between $M_{atm}$ and stream nitrate concentrations. When we plotted $M_{atm}$ as a function of the stream nitrate concentration ($[C_{total}]_{avg}$) together with our

data, however, the correlation coefficient ($R^2 = 0.63$) was poorer than the $M_{atm}/D_{atm}$ ratio ($R^2 = 0.92$) and $[C_{atm}]_{avg}$ ($R^2 = 0.80$) (Fig. 8). While $D_{atm}$ was the same between the sites studied by Rose et al. (2015a), $D_{atm}$ varied for the sites studied by Rose et al. (2015a) and the sites in this study. The $D_{atm}$ at site KJ, for instance, was about twice as much as that at the sites studied by Rose et al. (2015a). We concluded that normalising $M_{atm}$ by $D_{atm}$ is indispensable to use them as an index for the stage

of nitrogen saturation.

**4 Concluding remarks**

Using the [17]O-excess of nitrate as a tracer, we clarified that the major source of nitrate in stream water eluted from the studied forested catchments was nitrate in groundwater. The present results imply that nitrate in groundwater is the major source of nitrate in stream water eluted from forested catchments in humid temperate climates. Moreover, we clarified that the seasonal variation in the concentrations of soil water nitrate was buffered by groundwater. As a result, caution is needed to clarify the causes of seasonal variations in chemical/isotopic compositions of stream water because a time-lag from variations in soil water can be anticipated.

The export flux of unprocessed atmospheric nitrate relative to the entire deposition flux ($M_{atm}/D_{atm}$ ratio) showed a clear normal correlation with the flux-weighted average concentration of stream nitrate, not only in the forested catchments studied in this paper but also in all forested catchments studied using the [17]O-excess of nitrate as a tracer. As a result, reductions in the biological assimilation rates of nitrate in forested soils, rather than increased nitrification rates in forested soils, are largely responsible for the increase in stream nitrate concentration resulting from nitrogen saturation. Furthermore, in addition to the stream nitrate concentration, the export flux of unprocessed atmospheric nitrate relative to the entire deposition flux ($M_{atm}/D_{atm}$ ratio) in each forested catchment is applicable as a new index of nitrogen saturation. Further studies are needed for stream nitrate exported from various forested catchments around the world to verify the present results, using the [17]O-excess of nitrate as a tracer of the unprocessed atmospheric nitrate in stream nitrate.

Additionally, we should enhance accuracy and precision for both the flow rates (V in Eqs. (3) and (4)) and the deposition rates ($D_{atm}$) to estimate precise $M_{atm}/D_{atm}$ ratio in each catchment. While the errors associated with the $\Delta^{17}O$ values directly influences the errors associated with the $C_{atm}/C_{total}$ ratios and $M_{atm}/M_{total}$ ratios, their influences on $M_{atm}/D_{atm}$ ratios were minor. Rather, the errors associated with the flow rates and $D_{atm}$ had much larger impact on the $M_{atm}$, and $M_{atm}/D_{atm}$ ratios.

**Acknowledgments**

We thank anonymous reviewers for valuable remarks on an earlier version of this manuscript. The samples analyzed in this study were collected through the Long-term Monitoring of Transboundary Air Pollution and Acid Deposition by the Ministry of the Environment in Japan. We thank Ayaka
5   Ikegami for drawing colour altitude maps of the studied sites and Masanori Ito, Kosuke Ikeya, Koji Takahashi, Takuya Ohyama, Shuichi Hara, Toshiyuki Matsushita, Takanori Miyauchi, Yoshiumi Matsumoto, Rei Nakane, Lin Cheng, and the other present/past members of the Biogeochemistry Group at Nagoya University for their valuable support during this study. This work was supported by a Grant-in-Aid for Scientific Research from the Ministry of Education, Culture, Sports, Science,
10   and Technology of Japan under grant numbers 15H02804, 16K14308, 15K12187, 17H00780, 26241006, and 24651002.

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

**Table 1: The average $NO_3^-$ concentration in stream ($[C_{total}]_{avg}$; μmol $L^{-1}$), the average unprocessed $NO_{3\ atm}^-$ concentration in stream ($[C_{atm}]_{avg}$; μmol $L^{-1}$), the annual export flux of $NO_3^-$ per unit area of catchment ($M_{total}$; mmol $m^{-2}$ $yr^{-1}$), the annual export flux of $NO_{3\ atm}^-$ per unit area of catchment ($M_{atm}$; mmol $m^{-2}$ $yr^{-1}$), the annual average $M_{atm}/M_{total}$ ratios, and the deposition flux of $NO_{3\ atm}^-$ per unit area of catchment ($D_{atm}$; mmol $m^{-2}$ $yr^{-1}$) in the studied catchments.**

|  | *Site KJ* | *Site IJ1* | *Site IJ2* |
|---|---|---|---|
| $[C_{total}]_{avg}$ (μmol $L^{-1}$) | 58.4 | 24.4 | 17.1 |
| $[C_{atm}]_{avg}$ (μmol $L^{-1}$) | 3.26±0.59 | 1.39±0.25 | 0.56±0.07 |
| $M_{total}$ (mmol $m^{-2}$ $yr^{-1}$) | 76.4 | 50.1 | 35.1 |
| $M_{atm}$ (mmol $m^{-2}$ $yr^{-1}$) | 4.26±0.78 | 2.88±0.52 | 1.15±0.13 |
| $M_{atm}/M_{total}$ | 5.6±1.0% | 5.7±1.0% | 3.3±0.4% |
| $D_{atm}$ (mmol $m^{-2}$ $yr^{-1}$) | 45.6±4.6 | 44.5±4.4 | 44.5±4.4 |
| $M_{atm}/D_{atm}$ | 9.4±2.6% | 6.5±1.8% | 2.6±0.6% |

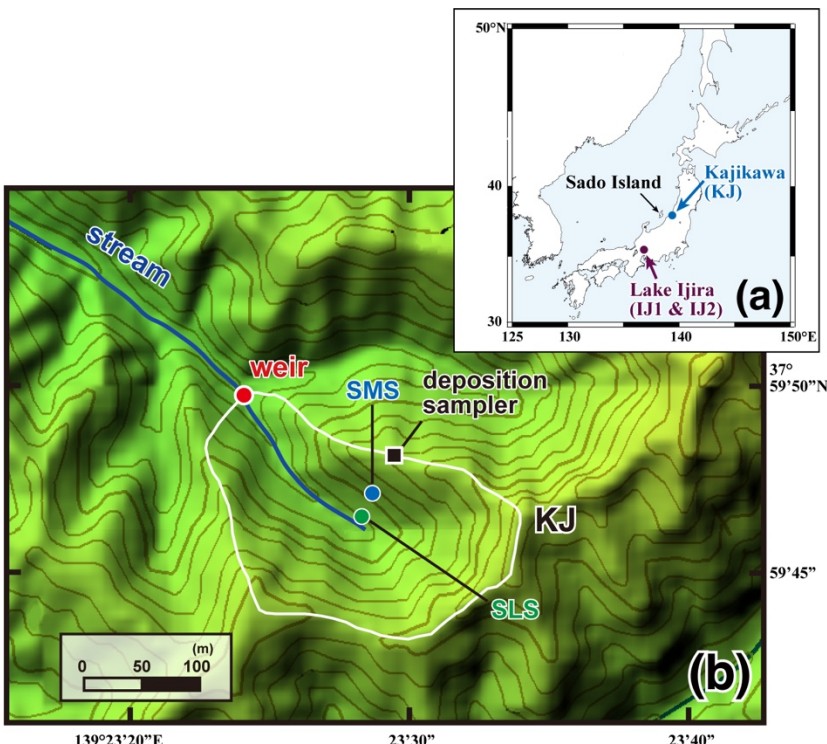

**Figure 1: A map showing the locations of the studied watersheds (Kajikawa and Lake Ijira) in Japan (a), and a colour altitude map of the site KJ (b), together with both the catchment area, shown by a white line, and the stream water sampling point, shown by a red circle (weir). The green and blue circles**
5 **denote the locations of soil water sampling (SLS and SMS, respectively), and the black square denotes the location where the deposition sampler was set.**

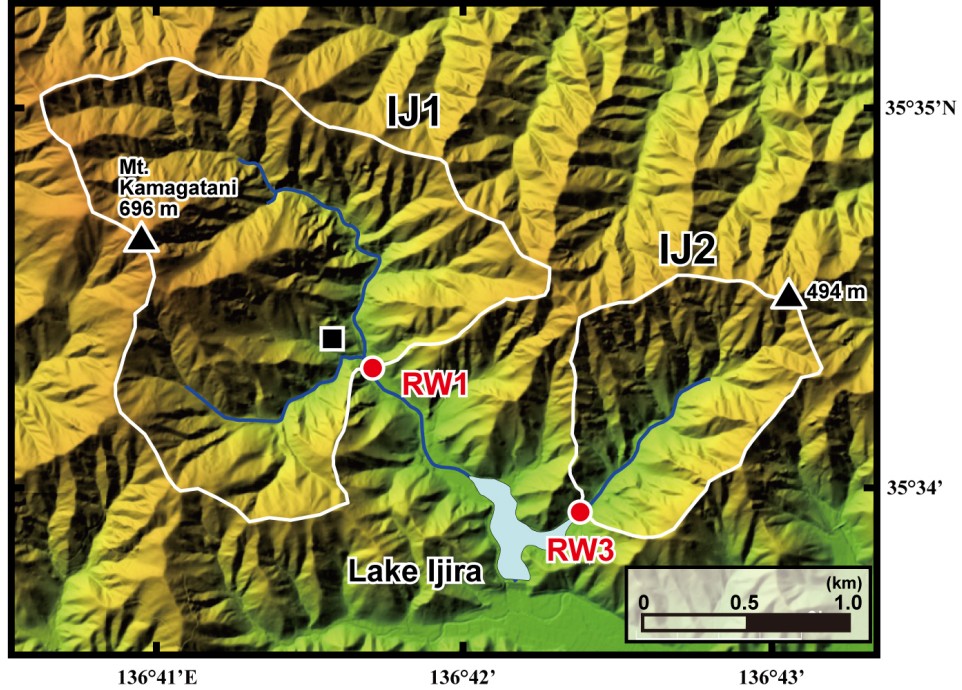

**Figure 2: A colour altitude map of the Lake Ijira watershed, together with the catchment areas, shown by a white line of the studied sites (IJ1 and IJ2) and the stream water sampling points, shown by red circles (RW1 for IJ1 and RW3 for IJ2). The black square denotes the location where the deposition sampler was set.**

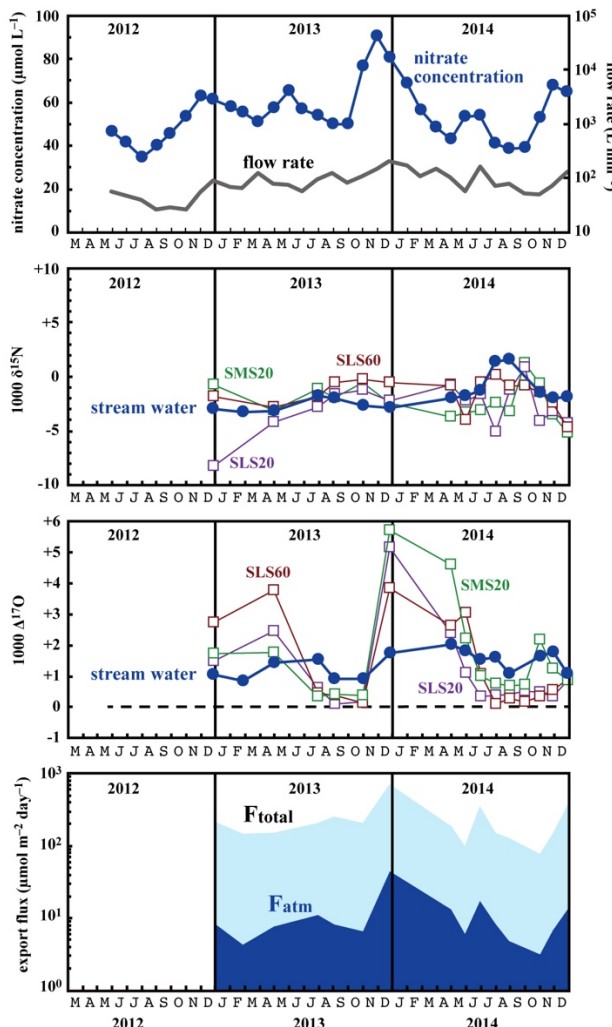

**Figure 3:** Temporal variations in the concentrations of nitrate (blue circles) and flow rates (grey line) in the stream water (a), together with those in the values of $\delta^{15}N$ (b), and $\Delta^{17}O$ (c) of the nitrate in stream water (blue circles) and soil water (SMS20: green squares, SLS20: purple squares, SLS60: brown squares), and in the export fluxes of nitrate ($F_{total}$) and atmospheric nitrate ($F_{atm}$) (d) via the stream at the site KJ.

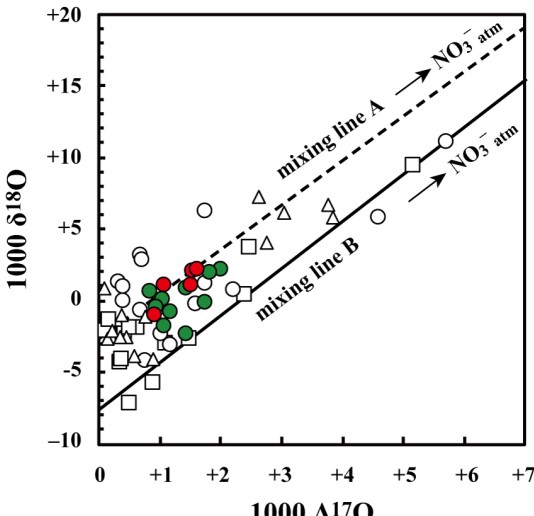

Figure 4: Relationship between $\Delta^{17}O$ and $\delta^{18}O$ values of nitrate in stream water at the site KJ (red circles: June, July, August, and September, green circles: rest of the months), together with those in soil water at the site KJ (SLS20: white squares, SLS60: white triangles, SMS20: white circles). A hypothetical mixing line between $NO_3^-{}_{atm}$ ($\Delta^{17}O=+26.3‰$, $\delta^{18}O=+79.8‰$; Tsunogai et al., 2016) and $NO_3^-{}_{re}$ having the average $\delta^{18}O$ value of $NO_3^-{}_{re}$ ($\Delta^{17}O=0‰$, $\delta^{18}O=-2.7‰$) in both stream and soil water in the site is shown (mixing line A), together with a hypothetical mixing between line between $NO_3^-{}_{atm}$ (the same $NO_3^-{}_{atm}$ with mixing line A) and $NO_3^-{}_{re}$ having the possible lowermost $\delta^{18}O$ value ($\Delta^{17}O=0‰$, $\delta^{18}O=-7.7‰$) that could be produced in the soils (mixing line B).

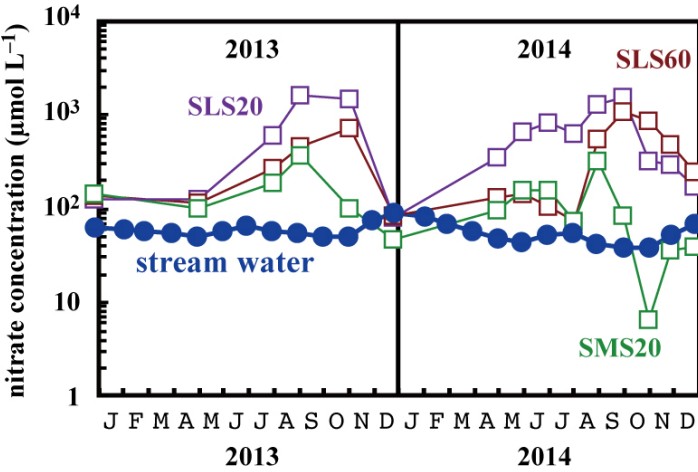

**Figure 5: Temporal variations in the concentrations of nitrate in stream water (blue circles) and those in soil water (SMS20: green squares, SLS20: purple squares, SLS60: brown squares) at the site KJ on a logarithmic scale.**

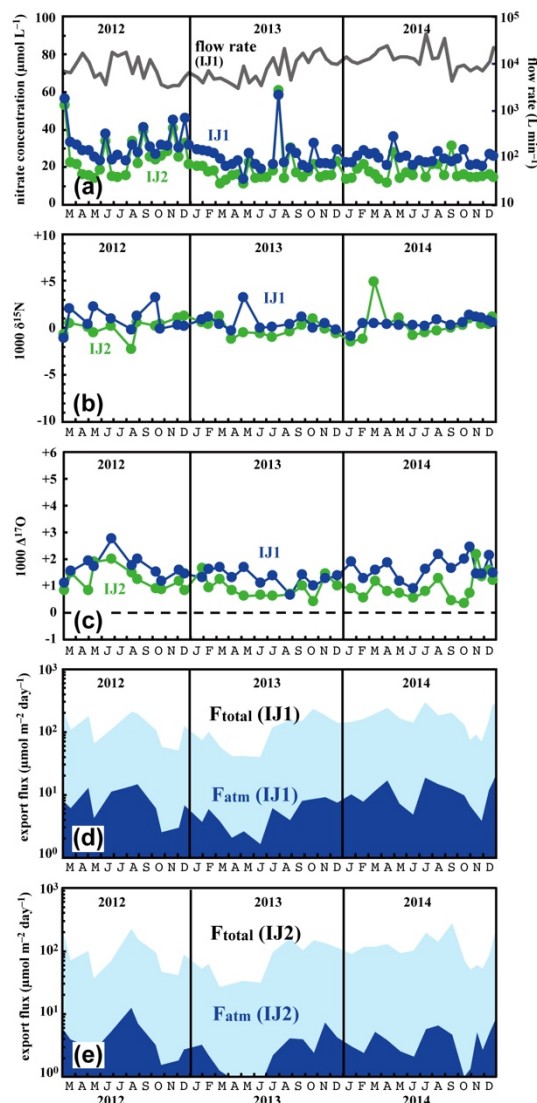

Figure 6: Temporal variations in concentrations of nitrate (IJ1: blue circles, IJ2: green circles) and flow rates at IJ1 (grey line) (a), together with those in the values of $\delta^{15}N$ (b), and $\Delta^{17}O$ (c) of nitrate at the sites IJ1 and IJ2, in the export fluxes of nitrate ($F_{total}$) and atmospheric nitrate ($F_{atm}$) via the stream at the site IJ1 (d), and in $F_{total}$ and $F_{atm}$ via the stream at the site IJ2 (e).

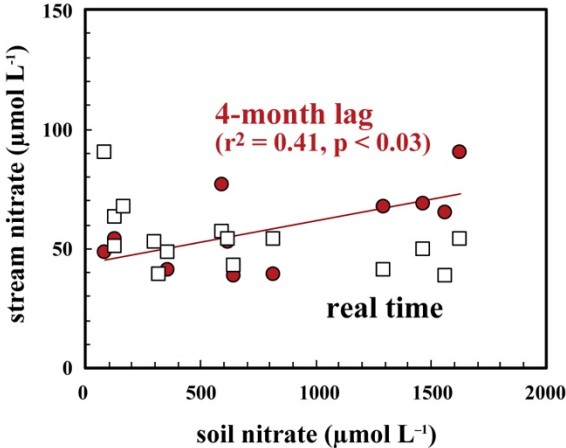

Figure 7: Relationship between concentrations of nitrate in soil water taken at SLS20 in site KJ and those in stream nitrate taken at the same time (white squares), together with those in stream nitrate taken 4 months later (red circles).

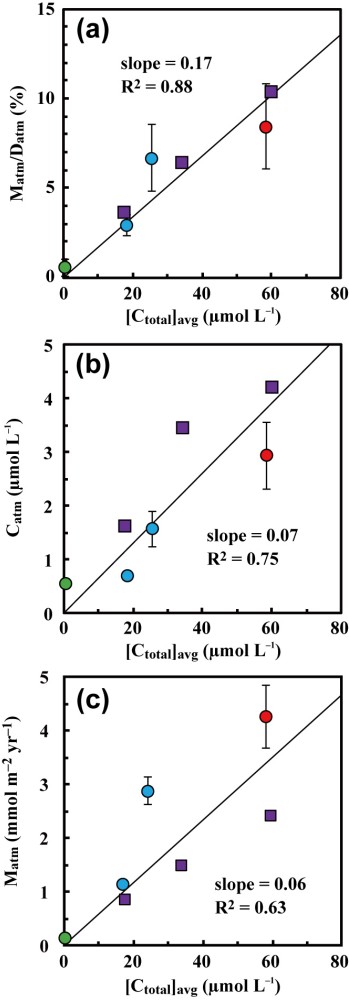

**Figure 8: The annual export flux of unprocessed $NO_3^-{}_{atm}$ relative to the annual deposition flux of $NO_3^-{}_{atm}$ ($M_{atm}/D_{atm}$ ratios) plotted as a function of the flux-weighted annual average concentration of nitrate in each stream water ($[C_{total}]_{avg}$) (a); the flux-weighted annual average concentration of $NO_3^-{}_{atm}$ in each stream water ($[C_{atm}]_{avg}$) plotted as a function of $[C_{total}]_{avg}$ (b); and the annual deposition flux of $NO_3^-{}_{atm}$ ($M_{atm}$) plotted as a function of $[C_{total}]_{avg}$ (c) (Site KJ: red circles, Sites IJ1 and IJ2: blue circles). Those determined at forested catchments in past studies are plotted as well, such as Fernow Experimental Forest in West Virginia, USA (purple squares; Rose et al., 2015a), and Teshio Experimental Forest in Hokkaido, Japan (a green circle; Tsunogai et al., 2014).**

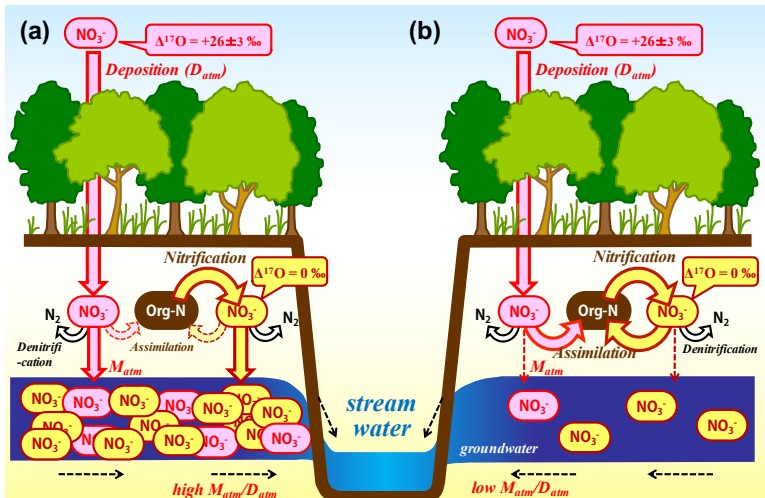

**Figure 9: Schematic diagram showing the biological processing of nitrate in a forested catchment under nitrogen saturation (a) and that under nitrogen-limited, normal forest (b) (modified after Nakagawa et al., 2013). All the arrows (=flows) related to the determination of the $M_{atm}/D_{atm}$ ratios are shown in red/pink, while the arrows (=flows) related to nitrification in soils are shown in brown/yellow.**

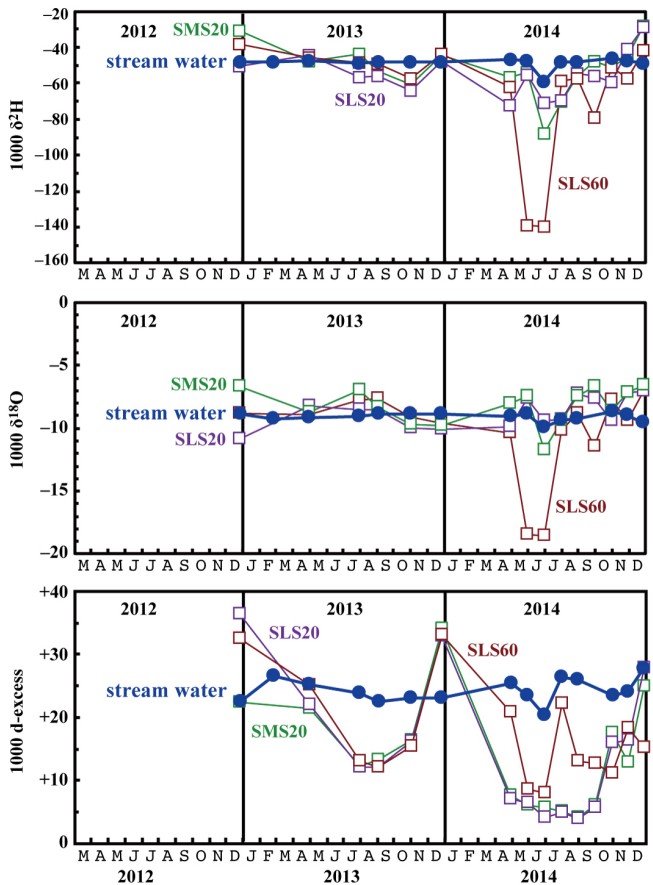

**Figure S1: Temporal variations in the values of $\delta^2H$ (a), $\delta^{18}O$ (b), and d-excess (c) of stream water (blue circles) and soil water (SMS20: green squares, SLS20: purple squares, SLS60: brown squares) at the site KJ.**

