# Peer review of "Export flux of unprocessed atmospheric nitrate from temperate forested catchments: A possible new index for nitrogen saturation"

_Biogeosciences, 2018_

## Referee Comment (RC1) · Anonymous Referee #1 · 2 Aug 2018

Overall Comment:

This manuscript by Nakagawa et al. present two-years of nitrate exports from three different forested catchments in Japan supposedly with different N saturation status. Using the triple isotope composition of nitrate in streams (3 catchewmnts) and soils (one catchment), they conclude on the sources of nitrate to the streams (atmospheric vs remineralized) and insist on the use of the Matm/Datm (atmospheric nitrate export flux/atmospheric nitrate deposition flux) ratio as an indicator of the N saturation status in the studied catchments. While I strongly advocate for a more systematic use of the triple isotope composition of nitrate to better understand N cycling in catchments, I find

this study lacking in several areas (see general and specific comments) and presents numerous flaws that, in my opinion, would require a lot more work before considered for publication. This would require more than major revisions to be amended, and believe that the manuscript in it current state should be rejected for publication.

General Comments: There are several points of concern that I would like to raise here:

First, the structure and presentation of the manuscript are not sound. There are several missing pieces of information in the Methods that are needed to correctly interpret the results presented here (see related specific comments). The number of nitrate isotopic data presented in Figures 3 and 6 do not correspond to the number of nitrate concentration, without any explanation as for why. A lengthy description of the results obtained for the catchments IJ1 and IJ2 is provided in the results section, but never discussed later on. The Figure 4 presents a line without any caption, nor associated equation. While the manuscript is understandable, a thorough English language editing will be needed, and should be favored before resubmitting the article.

Second, I am very concerned about the temporal resolution of the sampling performed in the study. While the authors say they sampled each stream about once a month (which I think is too low resolution to efficiently capture seasonal variations of atmospheric nitrate exports, especially during brief events like snowmelt than can occur in the course of two weeks (see Bourgeois et al., 2018)), the actual sampling frequency reported in Fig. 3 and 6 for nitrate isotopes is once every two months (7 data points per year). The authors should explain why this is so, and also justify that such a low temporal resolution is enough to capture the real seasonal variability of atmospheric nitrate exports.

Third, I would be more cautious regarding the simplification that high $NO_3^-$ concentration in a stream is always the result of N saturation. N saturation is a complex biogeochemical state of ecosystems of which one, among many, symptom is increased N leaching from soils to streams. But a high $NO_3^-$ concentration in streams can stem

from numerous other sources (e.g., topography, nature of the catchment soil/rock substrate, land-cover, percentage of forest cover) that need to be eliminated before the authors can indisputably corelate high NO3- concentration in streams and more advanced N saturation stage. Here, the authors conclude on the N saturation stage of the three studied catchments exclusively based on atmospheric nitrate exports from the streams. Not only is this not a novel finding (see next comment), but other evidences (e.g., nitrification/mineralization rates, leaves N content, roots and leaves mass) of the different N saturation stage between catchments should be provided to confirm, or not, the authors' conclusion.

Fourth, the main result of the manuscript, according to the authors, resides in using the Matm/Datm ratio as a new and robust indicator of N saturation status in forested catchment. I hardly find anything new in that result. As the authors point out in their discussion, the correlation between N saturation and increased export of atmospheric nitrate date as far back as two decades ago (Durka et al., 1994), and confirmed since then (Rose et al., 2015). Assuming that the gradient of nitrate concentration across streams is really due to different N saturation stages between catchments, then the correlation between Matm (export flux of atmospheric nitrate) and N saturation stage is not novel. The authors fail to demonstrate what the use of the Matm/Datm instead of just Matm is more valuable, and of scientific importance.

Specific Comments:

L.11: awkward use of the word "representative". I think what you want to say is "most important"

L.13: remove "receiving".

L.15-16: you need to remove "probably". It is well-documented that N deposition is responsible for N saturation in forested ecosystems.

L.23-25: Among the listed processes leading to nitrate removal should appear i) microbial immobilization (different from assimilation) and ii) nitrate leaching.

Page 3 L.5: you need to cite Kendall et al., 2007 here.

L.8-9: this is an awkward definition of unprocessed atmospheric nitrate. You need to find another word than "survives" here. A better definition would "atmospheric nitrate that has not undergone a full cycle of assimilation, mineralization and nitrification leading to the regeneration of nitrate, nor exchanged O atoms with H2O after deposition.".

L.12: This sentence should say ". . . we can quantify the proportion of unprocessed . . ."

L.2: The first sentence doesn't really make any sense. It should say: "The natural stable isotopic composition of nitrate is represented by its d15N, D17O, and d18O values."

L.5: You have introduced the NO3-atm notation for atmospheric nitrate in page 2 line 23. Please make a consistent use of that notation throughout the text, instead of alternatively using "atmospheric nitrate" and "NO3-atm"

L.8-10: you should specify here that remineralized nitrate also applies to atmospheric nitrate that has undergone a full cycle of assimilation, mineralization and nitrification. L.12: you say that the D17O of NO3-re is close to 0. This is very vague, please specify the range of D17O here, and/or what process are responsible for such value (different from 0).

L.15-17: I don't understand how you can conclude that based on the literature you provide. Only Tsunogai et al. (2016) presented a dataset of D17O-NO3-atm longer than a year (3 years), and none of these studies say that the annual average of D17O-NO3-atm is "almost constant" over time. Do you try to say the mean annual D17O-NO3-atm is similar in all these studies? They range in a similar array of values, but can still vary by a few ‰ depending on the geographic location (see Alexander et al.,

2009). This sentence needs to be much improved or utterly removed.

L.21: awkward use of "partial metabolism". Please rephrase.

L.25: To use this equation, you must assume that D17O- NO3-re = 0. This is not consistent with your statement line 12 that D17O- NO3-re is "close to 0".

L.7: I assume this is the associated error to the mean D17Oatm you are using in this study. How does this error translate in terms of uncertainty in your calculations of Catm, Fatm and Matm? I think this is an important, yet overlooked, piece that is missing in your manuscript. L.11: I am intrigued by how you choose your references: it is sufficient to cite the 3-4 works that first used this correction method (that would be Tsunogai et al., 2010 and 2011, Dejwakh et al. 2012 and Riha et al., 2014 I believe). Here it looks like you want to provide a list of all the works that used this method. This is neither necessary, nor actually accurate (i.e., you missed other works that also used it). L.25: Do you know the respective contribution of summer/winter precipitation to the annual total? If yes, please specify it here.

L. 1 and 14: Do you know the surface proportion of each watershed actually being covered by forests? If yes, please specify it.

L. 2: Since you are talking here of high loading rate, please provide the value.

L. 3: the use of "enrichment" instead of "concentration" throughout the manuscript is very confusing and will need to be amended.

L. 4: So, you assume that this catchment is N-saturated, according to Aber et al. (1998) definition of N-saturation. Please say it in these terms here.

Page 7 L.7-9: Please indicate how the sampling was conducted: manually, autosampler, what kind of bottles, cleaning procedures, etc.

L.13: I don't understand why you say "at each weir". Does it mean that the IJ1 catchment has several outlets where you sampled water? Please correct accordingly.

L.15-17: That is a pretty big assumption. It would seem to me that the topography of a catchment would impact the discharge rate far more than its area, because topography would drive both snow height in winter and water residence time the rest of the year. You should at the minimum provide some references to explain why you can make such assumption.

L.22: Please provide the proportion of samples collected during the winter period vs the rest of the year.

Section 2.4: I have several concerns regarding this section. First, regarding the extended amount of time you left the collecting bottle at the KJ site (around a month), and considering that in summer you have temperature as high as 34C (according to section 2.1), how can you be sure that your nitrate concentrations are not biased by partial evaporation of the rain water? Oppositely, do you have a heating system to melt the snow in winter? If not, how can you be sure that you really collect 100% of wintertime precipitation? Second, you state several factors that could impact the nitrate concentration in deposition samples (incomplete dry dep collects, possible nitrification). It would be useful that you provide an associated uncertainty to the estimated atmospheric NO3-atm concentration due to these factors (and also water evaporation). Third, you say that the deposition collector at the KJ site was installed in an open field. You must be aware that rainfall in open field is not representative of throughfall that actually reaches the soils and streams under forest canopies. For instance, Guerrieri et al. (2015) suggested that in forests with high N dep (which is the case at the KJ catchment) canopies play a significant role in modifying both NO3- concentration and isotopic composition from rainfall to throughfall. This is an important point that need to be clarified as it could impact the interpretation of your results (see below).

L.14: Does it mean that in the end, all samples were analyzed, and none rejected?

L.15-21: So, the values given in the manuscript are the averages of these repeated analyses?

L.26: Please describe what would be the highest uncertainty caused by presence of nitrite in a sample on the D17O value of nitrate (highest uncertainty would be for NO2- = 0.049 umol.L-1 and the lowest NO3- concentration you measured in your study). This would give the reader a better idea for why you regard nitrite concentrations as negligible.

L.10-14: It might be worth illustrating this by a figure that you could place in the SI, for readers unfamiliar with the different definitions of D17O and their discrepancies.

Page 12 L.6-12: A detail, but I don't think that Matm, Mre, Mtot and Datm can be classified as "fluxes", but more as "annual loads". L.13 Add "annual" before "deposition". L.23: I disagree, in 2014 the flux is still higher in spring than in December. You could probably replace "December" by "winter period". L.25: In 2015, you also have a June-July maximum in Fatm that is noticeable.

L.5: If I understood correctly, you calculated a Fatm value for every D17O value measured in the stream. Which means 15 points over two years, according to Figure 3. I think this temporal resolution is way too low to accurately catch the seasonality in stream atmospheric nitrate export (and for instance snowmelt in spring). How can you make a strong statement on this subject with seven samples per year? You really need to argue hear why such low sampling resolution is enough to describe the behavior of your catchment.

L.6-7: It would be useful to the reader if you could compare here your stream nitrate concentration with other concentration measured in forested catchments outside of Japan, just like you do for the isotopic composition of nitrate later on.

L.15: up to 1.6 mmol.L-1, or more? If it is more, then put the maximum value. As it is now, it does not make any sense.

L.23-27: This is interpretation of the results: therefore, it needs to be removed from the Results section and moved to the Discussion section.

Page 14 L.4-18: It is more standard to present deposition fluxes in kg-N ha-1 yr-1. This would be easier to compare with other studies and to understand how elevated deposition is on your catchments (and it is very elevated!). You may want to change your stream fluxes as well to be homogeneous on the units you use. L.17: Ok, so here is the error associated with your deposition estimation. Please refer to this section where appropriated earlier in your manuscript. L.21-22: Why don't you present also Fatm and Ftot for IJ2? If you don't present these results, then remove the section where you say that you extrapolated discharge data for IJ2 from IJ1 using the catchment area as converting factor.

Page 15 This is a very long description of the results observed at IJ 1 and 2 catchments, that are never discussed later in the discussion. Why is it so? The described patterns on this page look very interesting and to my opinion would deserve a thorough analysis later on! For instance, what causes the sporadic increase in nitrate concentrations in both streams (precipitation events?) and why is IJ1 more enriched in nitrate compared to IJ2 (more precipitation? Different percentage of land cover by forests?). L.23-26: This is very far stretched. Did you conduct a statistical test to verify the decreasing trend in concentration? Also, how can you say it started in 2000 when you report three years of data covering 2012-2014?

L.10: Why not giving the corresponding proportions of nitrate for each source using D17O to calculate the percentages.

L.17-20: This sentence doesn't make sense. How can D17O of soil nitrate reflect the original value of nitrate in groundwater? I think that soil nitrate reflects atmospheric nitrate D17O value, that is buffered by nitrification in soils (as shown by the seasonal variation in figure 3). Soil nitrate isotopic composition is not related to groundwater nitrate isotopic composition.

L.13: You need to indicate how you come up with this value. If you used the following equation: $\delta18O\text{-}NO{-}3 = 1/3\ (\delta18O\text{-}O2) + 2/3\ (\delta18O\text{-}H2O)$, then you must also comment on the limitations of its utilization (see Rose et al. 2015a and Snider et al. 2010). You should also try to plot d18O vs d15N and see how they correlate to confirm, or not, the absence of biological processes in your catchment.

L.17-19: This statement needs to be amended or removed. This is hardly new results, as the relationship between D17O and d18O have been used for almost a decade to understand biological processes in catchments (see previous work by Tsunogai et al.).

L.21-23: Is that an assumption or something you know for a fact? Please provide the data to justify that rain is responsible for Ftotal increase for winter (either the precipitation chart, or some data indicating that there is more precipitation in winter than I summer).

L.6: Add reference to Michalski et al. 2004 after the value of 0.

L.9: Here you should provide a scatterplot of d18O vs d15N to discuss the presence, or not, of any correlation between these isotopic values throughout the year. And even if you don't find any correlation, it does not mean that no assimilation is taking place (it would be really surprising to have no assimilation anywhere in the catchment) but rather

than the recharge of new NO3-re is overprinting isotopic fractionation by assimilation (Granger and Wankel, 2016).

L.18-19: Be more precise: you are talking about concentrations here. To say "stream nitrate shows a normal correlation with soil nitrate" doesn't make any sense. Also, I am a bit dubious of the strength of your correlation given the low number of samples presented in Figure 7 (n=11).

L.20-25: Alternatively, the slightly higher D17O values in winter/spring comparted to the rest of the year are due to freeze/thaw events leading to partial snowmelt, that is not well captured by your very coarse sampling resolution. How can you exclude that hypothesis, especially as you said that the KJ catchment is covered by snow from December to March, exactly when the Fatm is higher? This needs at least to be discussed. Didn't you measure water isotopes as well? Can't you tell from these measurements if the water comes from groundwater or from snowmelt (Hall et al., 2016; Liu et al., 2004)? That would be a very strong addition to your reasoning.

L.1-15: It would be nice here that you refer to a table where you list the annual mean values for Catm, Ctot, Fatm, Ftot, and Matm for each watershed. It is hard to keep up with all you say because we are always looking for the values elsewhere in the manuscript.

L.16: There is no Equation (9) in your manuscript

L.17: You need to detail how you calculate your incertitude either in the Methods or in the SI: you stipulated earlier in the manuscript that you would assume a 20% error on the Datm at the KJ catchment (see Page 14, L.17). Can that result in the 2.6% error on the Matm/Datm ratio that you present here? Did you perform a formal error propagation calculation? Please expand more on this aspect. Also, I would be very curious on how you obtained your percentages. If I divide the Matm (=8.8, 5.7, and

2.2 for KJ, IJ1 and IJ2 respectively) by Datm (=45.6, 49.2, and 48.3 for KJ, IJ1 and IJ2 respectively), I obtain 19.3, 11.6 and 4.8%. Not at all what you calculated. Please explain

L.1: That is something that bothers me in your manuscript: the link you draw between catchment N saturation and stream nitrate concentration seems very hazardous to me. Increase of stream nitrate concentration can be a symptom of a higher N saturation status, but N saturation is not per se the only reason that could explain higher N concentration in a stream (topology, geomorphology, land-cover are other very strong factors that can influence N exports in streams). You need to explain more why it is the N saturation status that drive higher nitrate export from the KJ site compared to the IJ catchments, and not the percentage of forest cover on the catchment for instance, or another parameter (like snow pack height).

L.7-8: I don't understand what in Figure 9 can lead to such conclusion: I am pretty sure that contrary to what you say, different nitrification rates in soils will lead to different NO3-re leaching fluxes to stream/groundwater, and thus impact the D17O value of nitrate in stream (by dilution). This would impact Matm, and therefore the Matm/Datm ratio. So please clarify what you meant here.

L.13-22: To me, you really fail to demonstrate here what your ratio (Matm/Datm) brings more in term of N saturation understanding than just the use of Matm, which was already described in previous studies. I don't see anything new here.

L.10-13: That is my point. Rose et al 2015 already showed the relationship between Matm and N saturation? So, what is new here? What does your ratio bring more than just the use of Matm? Also, please try to plot the same correlation with Matm instead of Catm, and report the correlation strength.

Figure 3: Why is there not the same number of samples for nitrate concentration and isotopic values for the stream? The temporal resolution of nitrate isotopes as presented in this graph is very low (one sample every two months) and not enough to capture seasonal events such as snowmelt. This needs to be clarified as it could substantially change the results interpretation and the overall study conclusion.

Figure 4: What is the line in black? Is it the regression line, and if yes, of what (only streams, streams + soils)? Add slope and p-value. If it is the mixing line, say it in the caption and show the two end-members (atmospheric nitrate and remineralized nitrate)

Figure 5: Same question as for Figure 3.

Figure 8: Please add a third panel to show Matm vs Ctot.

Bourgeois, I., Savarino, J., Caillon, N., Angot, H., Barbero, A., Delbart, F., Voisin, D. and Clément, J.-C.: Tracing the Fate of Atmospheric Nitrate in a Subalpine Watershed Using Δ17O, Environ. Sci. Technol., 52(10), 5561–5570, doi:10.1021/acs.est.7b02395, 2018. Durka, W., Schulze, E.-D., Gebauer, G. and Voerkeliust, S.: Effects of forest decline on uptake and leaching of deposited nitrate determined from 15N and 18O measurements, Nature, 372(6508), 765–767, doi:10.1038/372765a0, 1994. Granger, J. and Wankel, S. D.: Isotopic overprinting of nitrification on denitrification as a ubiquitous and unifying feature of environmental nitrogen cycling, Proceedings of the National Academy of Sciences, 113(42), E6391–E6400, doi:10.1073/pnas.1601383113, 2016. Hall, S. J., Weintraub, S. R., Eiriksson, D., Brooks, P. D., Baker, M. A., Bowen, G. J. and Bowling, D. R.: Stream Nitrogen Inputs Reflect Groundwater Across a Snowmelt-Dominated Montane to Urban Watershed, Environmental Science & Technology, 50(3), 1137–1146, doi:10.1021/acs.est.5b04805, 2016. Liu, F., Williams, M. W. and Caine, N.: Source waters and flow paths in an alpine catchment, Colorado Front Range, United States: SOURCE WATERS AND FLOW PATHS IN ALPINE CATCHMENTS, Water Resources Research, 40(9), doi:10.1029/2004WR003076, 2004. Rose, L. A., Elliott, E. M. and Adams, M. B.: Triple Nitrate Isotopes Indicate Differing Nitrate Source

Contributions to Streams Across a Nitrogen Saturation Gradient, Ecosystems, 18(7), 1209–1223, doi:10.1007/s10021-015-9891-8, 2015.

---

## Referee Comment (RC2) · Anonymous Referee #2 · 4 Aug 2018

This manuscript is worth publication in Biogeosciences even though some major revisions are necessary. The paper presents temporal variations NO3 concentrations and isotopic signatures (15N, 18O, 17O) in streams and soils of three forested watersheds in Japan. Stream discharge rates and total atmospheric NO3 deposition rates were also measured and used to calculate the daily and annual watersheds export fluxes of nitrate (total, atmospheric, and "mineralized"). Using concentrations and ïĄĎ17O, the authors show that there is a 4-months lag-time for NO3 originating from soil nitrification to reach the stream through groundwater transportation. They found that the proportion of atmospheric NO3 deposited (Matm/Datm) that leaves the watershed was positively related to the amount of total nitrate exported by the streams, which I think is

rather intuitive. This said, they demonstrate that nitrate loss in N-saturated catchments was due to a reduction of its assimilation by plant and microbes. Finally, they suggest that the ratio Matm/Datm (or the Catm in stream water) could be used as an indicator of watershed nitrogen saturation particularly useful in humid climate. Although I think the paper brings new and interesting data, I identify three main points that challenge how much we can trust the conclusions. First, the watersheds (KJ, IJ1 and IJ2) have extremely different size (3.84, 298, 108 ha respectively) and there are no information concerning their geology and soil depth/quality which are important parameters to interpret stream discharge and nutrient flow. Second, stream discharge is estimated with a different method for each stream (section 2.2). Third, atmospheric nitrate deposition rates were also obtained with different approaches in the three watersheds (section 2.4). Finally one can wonder how much data variation is carried by each of these points and how this influences the results and their interpretation. My guess is that this should be at least acknowledged, analysed and discussed thoroughly to convince the reader.

Abstract: L21: I suggest you give the annual fluxes of nitrate instead of concentrations.

L23-25: It is misleading to compare, in the same sentence, 17O-excess in KJ soil water with those of KJ, IJ1 and IJ2 stream water. Add "in KJ" after "+0.1‰ to+5.7‰'.

L25:"was groundwater nitrate". I would also remove the end of the sentence "which buffered the seasonal. . ." as it is confusing.

Introduction:

P2-L5 (and throughout the paper): Do not use the word "enrichment" when you simply mean "increase", especially in a paper dealing with isotopes for which "enrichment" is usually used as in P4-L7.

P2-L25: There are other processes that should be mentioned : DNRA for instance.

P3-L2-3: Remove the end of the sentence ", including [. . .] catchments.", it does not

bring new information.

P3-L4: composition. . .has been (no plural)

P3-L8: "has survived"

P3-L13: remove "nitrate including"

P3-L21-22: this sentence is useless.

I miss some clear hypotheses at the end of the introduction.

Section 1.2 is Materials and Method.

P4-L6: atmospheric nitrate is not only produced by photochemical reactions.

P4-L13: specify this is Equation 1

P4-L24: Specify this is Equation 2

Here you use Catm and Ctotal for the first time. In the rest of the manuscript they represent either mean daily or mean annual concentrations which is confusing. Please use different codes for daily and annual.

Experimental section

KJ, IJ1 and IJ2 have very different size. Here (and probably in the discussion) you should somehow reassure the reader that it is not a source of bias in your interpretation.

P5-L15: Remove "continuous" or replace by "weekly" as samples were collected once a week.

P5-L25: "2500 mm" seems a lot as the average precipitation in Shibata is 1263 mm.

P6-L9: "3300 mm" seems a lot. Actually if I do the maths with your figures I get 1500 mm for 10 months/yr.

P6-L23: I am not sure Clethra sp. and Ilex sp. can be considered as trees.

P7-L1: You do not use water T°, pH, alkalinity and EC in the paper. Please remove. In fact, maybe it could bring new insights in your data set and interpretation.

Section 2.2: Discharge is estimated with different methods for each stream. This is a source of variation for your results and you should acknowledge that somewhere in the discussion to convince the reader you took this issue into account in your interpretation. There is the same risk of discrepancy with the way atmospheric nitrate deposition rates were obtained (section 2.4).

Section 2.3: what SLS and SMS stand for? You need to explain why you decided to sample next to the stream and 20m upland, and why at 20cm and 60cm depth.

P7-L22: "between December and March" Section 2.5: I am surprise samples were not acidified prior storage.

P9-L15: remove ", the procedure [. . .]. Approximately"

P10-L22: You mention NO2 as a possible source of variation, what about NH4? Section 2.7: I suggest you give the units for each Equation.

P12-L1: replace "obtain" by "estimate"

Results

P12-L16-21: You can remove this paragraph which does not bring new information compared to the figures legends.

P12-L25: Could you add ranges or values to Ftotal and Fatm. Same thing for IJ1 and IJ2.

P14-L20-22: You can remove this sentence which does not bring new information compared to the figures legends.

Discussion

Specify to which category (catchment groundwater, through flow) belongs the water

sampled in SLS and SMS.

P16-L25: Add to Fig 4 legend that the hypothetical mixing line is reported.

P17-L1: replace "buffered by" by "diluted in"

P18-L22: "This delay time reflects the magnitude and flow of the nitrate..."

P19-L23-24: A figure showing the relationship between Matm and stream NO3 conc. would be welcome.

P20-L3: replace "N" by "NO3"

P20-L4: replace "nitrogen" by "nitrate".

P20-L8: Fig. 9 does not show that "Matm/Datm ratios are stable during the progress of nitrification in forested soils"

P20-L17-22: this paragraph is not clear enough. In particular the last sentence compare "stream nitrate enrichment due to nitrogen saturation" with "stream nitrate enrichment due to artificial processes", which I do not quite understand. Artificial processes (e.g. fertilizers, leguminous fields...) are responsible for N saturation. Please clarify.

P21-L24: "...by 6 and 20 times respectively in accordance..."

Fig.2: Underneath RW1 and RW3 there are distances (120m and 40m respectively), are they elevation a.s.l.?

Fig1 and Fig.2: Please use the same kind of map for both figures.

Fig.4: Please specify what the line stands for?

---

## Author Comment (AC1) · 16 Sep 2018

Reply to your comment (Referee #1).

Thank you very much for your valuable comments on our manuscript. We have responded to each of your comments and questions.

> The number of nitrate isotopic data presented in Figures 3 and 6 do not correspond to the number of nitrate concentration, without any explanation as for why. > Figure 3: Why is there not the same number of samples for nitrate concentration and isotopic values for the stream? The temporal resolution of nitrate isotopes as presented in this

graph is very low (one sample every two months) and not enough to capture seasonal events such as snowmelt. This needs to be clarified as it could substantially change the results interpretation and the overall study conclusion. > Figure 5: Same question as for Figure 3.

The samples presented in this study were collected through a project (the Long-term Monitoring of Transboundary Air Pollution and Acid Deposition) promoted by the Ministry of the Environment in Japan. The flow rates of the streams and concentrations of nitrate and other constituents were determined through the project as well. On the other hand, the measurements on the stable isotopes of nitrate were not included in the project, because we were unable to determine the stable isotopes of nitrate at the beginning of the project in 2003. The stable isotope analysis was done for the archived samples, in support of a different project in 2014. Because the archived samples were precious and the measurements of the $\Delta 17O$ values of nitrate were costly and time consuming, the number of samples for stable isotopes were limited to 1/2 of the whole at site KJ. We will clarify this in the revised MS. Despite this, 134 $\Delta 17O$ values are reported in this MS. We repeated the analysis for each sample at least three times to attain high precision (see section 2.5). We hope that our results, including these many data values, is worthy of publication.

We have addressed your concerns about snowmelt in a later reply.

> A lengthy description of the results obtained for the catchments IJ1 and IJ2 is provided in the results section, but never discussed later on. > Page 15 This is a very long description of the results observed at IJ1 and 2 catchments, that are never discussed later in the discussion. Why is it so? The described patterns on this page look very interesting and to my opinion would deserve a thorough analysis later on! For instance, what causes the sporadic increase in nitrate concentrations in both streams (precipitation events?) and why is IJ1 more enriched in nitrate compared to IJ2 (more precipitation? Different percentage of land cover by forests?).

The data obtained at catchments IJ1 and IJ2 were integrated into Figure 8, the most important figure for discussion in this manuscript. We think the current length of this section is inevitable if the data obtained at catchments IJ1 and IJ2 are to be justified and the data is to be integrated into Figure 8.

Clarifying the reason for the sporadic increases detected at IJ1 and IJ2 was not the objective of this study. While the present results imply that the sporadic increases did not accompany significant changes in $\Delta$17O (see figure 6), a much higher time resolution of the samples is required to verify the results. Because the sampling interval was set to once a month under the project (Long-term Monitoring of Transboundary Air Pollution and Acid Deposition), an investigation of the reason for the sporadic increases was not possible in this study, and should be reserved for future study. The reason for the differences between the data from IJ1 and that from IJ2 is discussed later, in section 4.3.

> The Figure 4 presents a line without any caption, nor associated equation. > Figure 4: What is the line in black? Is it the regression line, and if yes, of what (only streams, streams + soils)? Add slope and p-value. If it is the mixing line, say it in the caption and show the two end-members (atmospheric nitrate and remineralized nitrate)

That is the regression line for both streams and soils. We will clarify this in the figure caption of the revised MS. We have added the p-value as well. Additionally, we will add the mixing line between the two end-members (atmospheric nitrate and remineralized nitrate) in the figure during the revision.

> While the manuscript is understandable, a thorough English language editing will be needed, and should be favored before resubmitting the article.

The English of the manuscript was thoroughly edited by Editage English editing service (http:// www.editage.jp/) prior to the submission. We intend to have them edit the English again prior to submission of the revised manuscript.

> Second, I am very concerned about the temporal resolution of the sampling performed in the study. While the authors say they sampled each stream about once a month (which I think is too low resolution to efficiently capture seasonal variations of atmospheric nitrate exports, especially during brief events like snowmelt than can occur in the course of two weeks (see Bourgeois et al., 2018)), the actual sampling frequency reported in Fig. 3 and 6 for nitrate isotopes is once every two months (7 data points per year). The authors should explain why this is so, and also justify that such a low temporal resolution is enough to capture the real seasonal variability of atmospheric nitrate exports.

The sampling for nitrate concentrations was performed once per month, while the samples for isotopes were selected from archived samples that had previously been measured for concentrations. Due to this reason, the number of samples for stable isotopes was limited; we have clarified this in the manuscript. While the total number of data points were 15 for nitrate isotopes during the 2 years of observation at site KJ, the most nitrate-enriched sample (Dec., 2013) and the second most nitrate-depleted sample (Oct., 2014) were included in the $\Delta17O$ analysis. Nevertheless, the nitrate isotopes showed little temporal variation with respect to the variation in nitrate concentration, and the variation range (1 sigma) was less than 0.4‰ for $\Delta17O$. Without $\Delta17O$ data it is difficult to assume sudden changes only in $\Delta17O$ when there are no sudden changes in nitrate concentration during the period. Additionally, our results suggest that the seasonal variabilities in both atmospheric nitrate and soil nitrate were buffered by groundwater nitrate at the site, as discussed in section 4.1. We concluded that the total number of data points (15) was significant for determining the average isotopic compositions ($\Delta17O$ of nitrate, especially) at the site.

> Third, I would be more cautious regarding the simplification that high NO3- concentration in a stream is always the result of N saturation. N saturation is a complex biogeochemical state of ecosystems of which one, among many, symptom is increased N leaching from soils to streams. But a high NO3- concentration in streams can stem

from numerous other sources (e.g., topography, nature of the catchment soil/rock substrate, land-cover, percentage of forest cover) that need to be eliminated before the authors can indisputably corelate high NO3- concentration in streams and more advanced N saturation stage. Here, the authors conclude on the N saturation stage of the three studied catchments exclusively based on atmospheric nitrate exports from the streams. Not only is this not a novel finding (see next comment), but other evidences (e.g., nitrification/mineralization rates, leaves N content, roots and leaves mass) of the different N saturation stage between catchments should be provided to confirm, or not, the authors' conclusion. > Page 20/L.1: That is something that bothers me in your manuscript: the link you draw between catchment N saturation and stream nitrate concentration seems very hazardous to me. Increase of stream nitrate concentration can be a symptom of a higher N saturation status, but N saturation is not per se the only reason that could explain higher N concentration in a stream (topology, geomorphology, land-cover are other very strong factors that can influence N exports in streams). You need to explain more why it is the N saturation status that drive higher nitrate export from the KJ site compared to the IJ catchments, and not the percentage of forest cover on the catchment for instance, or another parameter (like snow pack height).

There have previously been many ecological and biogeochemical studies on the high elution rates of nitrate (site KJ: Kamisako et al., 2008, Sase et al., 2008; 2012, IJ1 site: Yamada et al., 2007; Nakahara et al., 2010). The possible factors that you suggested (topography, nature of the catchment soil/rock substrate, land-cover, percentage of forest cover, etc.) have been considered in previous studies as possible reasons for the high elution rates of nitrate. However, Kamisako et al. (2008) proposed that site KJ was at nitrogen saturation, probably due to the excess loading of nitrogen from the atmosphere. Nakahara et al. (2010) proposed that site IJ1 was at nitrogen saturation (stage 2) as well. While their studies served as important backgrounds of our research, we don't think it is worthwhile to repeat their discussions in our manuscript. Instead,we will emphasize that our conclusion regarding the nitrogen saturation at the studied sites agrees with those of these previous studies.

> Fourth, the main result of the manuscript, according to the authors, resides in using the Matm/Datm ratio as a new and robust indicator of N saturation status in forested catchment. I hardly find anything new in that result. As the authors point out in their discussion, the correlation between N saturation and increased export of atmospheric nitrate date as far back as two decades ago (Durka et al., 1994), and confirmed since then (Rose et al., 2015). Assuming that the gradient of nitrate concentration across streams is really due to different N saturation stages between catchments, then the correlation between Matm (export flux of atmospheric nitrate) and N saturation stage is not novel. The authors fail to demonstrate what the use of the Matm/Datm instead of just Matm is more valuable, and of scientific importance. > Page 20/L.13-22: To me, you really fail to demonstrate here what your ratio (Matm/Datm) brings more in term of N saturation understanding than just the use of Matm, which was already described in previous studies. I don't see anything new here.

We did not indicate, in any part of this paper, that the high Matm/Datm (or Matm) in an N saturated forest was a new finding of this study. Based on the results reported in previous studies, such as those that you pointed out (Durka et al. 1994; Rose et al., 2015), as well as those of our own previous studies (Tsunogai et al., 2010; 2014; 2016), we expected (1) that the biological metabolic processes of nitrate in forest soils primary will control the Matm/Datm ratios of nitrate, and (2) that the Matm/Datm ratios will increase with increase in the concentration of stream nitrate (i.e., elution rate of nitrate). We verified these expectations in this study.

The Matm/Datm ratio, the directly exported atmospheric nitrate flux relative to whole deposition flux of atmospheric nitrate in a catchment area, was used in our previous study as an index to evaluate the biological metabolic rate of nitrate in forest soils in a catchment (Fig. 9) (Tsunogai et al., 2014). Because the metabolic rates of nitrate (almost equal to the biological assimilation rates of nitrate) in forest soils primarily determine the (Datm – Matm)/Datm ratio (removal ratio of atmospheric nitrate to the total atmospheric nitrate deposited in a catchment; Tsunogai et al., 2014), using Matm/Datm

ratio as the index, instead of the Matm, is essential in principle. Also, because Datm is variable between the study sites (Datm at site KJ is about twice as much as that at the site studied by Rose et al. (2015)), normalizing Matm by Datm is indispensable. We will emphasize this in the revised MS. Furthermore, we will present the relationship between average nitrate concentration and Matm in the new version of Fig. 8.

> Specific Comments: > Page 2/L.11: awkward use of the word "representative". I think what you want to say is "most important"

We will make the suggested revision.

> Page 2/L.13: remove "receiving".

We will make the suggested revision.

> Page 2/L.15-16: you need to remove "probably". It is well-documented that N deposition is responsible for N saturation in forested ecosystems.

We will make the suggested revision.

> Page 2/L.23-25: Among the listed processes leading to nitrate removal should appear i) microbial immobilization (different from assimilation) and ii) nitrate leaching.

We are very sorry but we cannot understand the meaning of the phrase "microbial immobilization (of nitrate) different from assimilation". Nonetheless, the aim of this sentence was to list the major processes in the catchment that control the nitrate concentration in the stream water, particularly those that influence the long-term changes in stream nitrate concentration. We think that most of the major processes were already included in the sentence.

> Page 3/L.5: you need to cite Kendall et al., 2007 here.

We will make the suggested revision.

Kendall, C., E. M. Elliott, and S. D. Wankel (2007) Tracing anthropogenic inputs of

nitrogen to ecosystems, in Stable Isotopes in Ecology and Environmental Science, 2nd edition, edited by R. H. Michener and K. Lajtha, pp. 375-449, Blackwell Publishing.

> Page 3/L.8-9: this is an awkward definition of unprocessed atmospheric nitrate. You need to find another word than "survives" here. A better definition would "atmospheric nitrate that has not undergone a full cycle of assimilation, mineralization and nitrification leading to the regeneration of nitrate, nor exchanged O atoms with H2O after deposition.".

We will make the suggested revision. The atmospheric nitrate involved in a part of the N cycle, however, no longer remained atmospheric nitrate; thus, the "full cycle" that you recommended is not necessary. We have changed the definition to "atmospheric nitrate that was supplied via atmospheric deposition and was not involved in the N cycle during the biological processing of nitrate, such as . . .".

> Page 3/L.12: This sentence should say ". . . we can quantify the proportion of unprocessed . . ."

The proportion of unprocessed atmospheric nitrate in total nitrate can be quantified only from $\triangle$17O. What we wanted to emphasize here was that we can quantify unprocessed atmospheric nitrate (we can determine the absolute concentration of unprocessed atmospheric nitrate) from $\triangle$17O and the concentration of stream nitrate. So as not to mislead readers, we will revise this as mentioned.

> Page 4/L.2: The first sentence doesn't really make any sense. It should say: "The natural stable isotopic composition of nitrate is represented by its d15N, D17O, and d18O values."

We will make the suggested revision.

> Page 4/L.5: You have introduced the NO3-atm notation for atmospheric nitrate in page 2 line 23. Please make a consistent use of that notation throughout the text, instead of alternatively using "atmospheric nitrate" and "NO3-atm"

We will make the suggested revision.

> Page 4/L.8-10: you should specify here that remineralized nitrate also applies to atmospheric nitrate that has undergone a full cycle of assimilation, mineralization and nitrification.

We will make the suggested revision.

> Page 4/L.12: you say that the D17O of NO3-re is close to 0. This is very vague, please specify the range of D17O here, and/or what process are responsible for such value (different from 0). > Page 4/L.25: To use this equation, you must assume that D17O- NO3-re = 0. This is not consistent with your statement line 12 that D17O- NO3-re is "close to 0".

We would like to use 0‰ for $\Delta$17O of NO3–re while citing a reference.

> Page 4/L.15-17: I don't understand how you can conclude that based on the literature you provide. Only Tsunogai et al. (2016) presented a dataset of D17O-NO3-atm longer than a year (3 years), and none of these studies say that the annual average of D17ONO3- atm is "almost constant" over time. Do you try to say the mean annual D17ONO3- atm is similar in all these studies? They range in a similar array of values, but can still vary by a few ‰ depending on the geographic location (see Alexander et al., 2009). This sentence needs to be much improved or utterly removed.

What we wanted to say was that the geographical difference in the annual average $\Delta$17O values of NO3–atm was less than a few ‰ in mid-latitude. We will revise this.

> Page 4/L.21: awkward use of "partial metabolism". Please rephrase.

We will rephrase this.

> Page 5/L.7: I assume this is the associated error to the mean D17Oatm you are using in this study. How does this error translate in terms of uncertainty in your calculations of Catm, Fatm and Matm? I think this is an important, yet overlooked, piece that is

missing in your manuscript.

The error in the $\Delta 17O$ value of NO3–atmÂă($\pm 3$‰, together with the error in the $\Delta 17O$ value of the sample nitrate in soils/stream ($\pm 0.1$‰ first translate into Catm via Eq. (2), based on the general propagation law of errors, and then translate into Fatm followed by Matm. Because the error in the $\Delta 17O$ value of NO3–atm (3‰ corresponds to 12% of $\Delta 17Oatm$ (+26‰ and the relative errors in the $\Delta 17O$ values of the sample nitrate were similar (around 10 %), the values of Catm, Fatm, and Matm showed errors of around 20% for each value. We did not overlook the errors but included them in the complete calculation processes presented in the text (e.g., page 19) and the figures (Fig. 8).

> Page 5/L.11: I am intrigued by how you choose your references: it is sufficient to cite the 3-4 works that first used this correction method (that would be Tsunogai et al., 2010 and 2011, Dejwakh et al. 2012 and Riha et al., 2014 I believe). Here it looks like you want to provide a list of all the works that used this method. This is neither necessary, nor actually accurate (i.e., you missed other works that also used it).

We will make the suggested revision.

> Page 5/L.25: Do you know the respective contribution of summer/winter precipitation to the annual total? If yes, please specify it here.

We will make the suggested revision.

> Page 6/L. 1 and 14: Do you know the surface proportion of each watershed actually being covered by forests? If yes, please specify it.

Except for the stream surface, the entirety of the catchment areas were covered by forests. We have clarified this in the revised MS, in the first sentence of section 2.1. You can see the area covered by forests on Google maps as well (while you can see deforestation along the stream at site IJ2 on Google maps, the deforestation occurred in the winter of 2015 so this did not interfere with our results).

IJ1: https://www.google.co.jp/maps/@37.9960808,139.3904356,403m/data=!3m1!1e3
RW1 and RW3: https://www.google.co.jp/maps/@35.5699514,136.6930194,4051m/data=!3m1!1e3

> Page 6/L. 2: Since you are talking here of high loading rate, please provide the value.

We will make the suggested revision.

> Page 6/L. 3: the use of "enrichment" instead of "concentration" throughout the manuscript is very confusing and will need to be amended.

We will make the suggested revision.

> Page 6/L. 4: So, you assume that this catchment is N-saturated, according to Aber et al. (1998) definition of N-saturation. Please say it in these terms here.

While Kamisako et al. (2008) proposed N-saturation at site KJ, we did not assume that the high nitrate concentrations in the studied streams were the result of N saturation, prior to obtaining the data. Because Japanese forests do not present seasonal variation in the elution rate of nitrate irrespective to the stage of N-saturation, it was difficult for us to assume that the site was N-saturated, according to the definition of N-saturation proposed by Aber et al. (1998). We studied site KJ because the stream showed high nitrate concentration while the catchment was fully covered by forest.

> Page 7/L.7-9: Please indicate how the sampling was conducted: manually, autosampler, what kind of bottles, cleaning procedures, etc.

The stream water samples were collected manually in bottles which were rinsed at least twice with the sample itself. In this study, 1L or 2L polyethylene bottles, washed using chemical detergents, rinsed at least thrice using deionized water, and then dried in laboratory, were used for collecting samples. We will add this in the MS in response to your request.

> Page 7/L.13: I don't understand why you say "at each weir". Does it mean that the IJ1 catchment has several outlets where you sampled water? Please correct accordingly.

This was a mistype, sorry. There was only one outlet at the IJ1 catchment. We have revised this.

> Page 7/L.15-17: That is a pretty big assumption. It would seem to me that the topography of a catchment would impact the discharge rate far more than its area, because topography would drive both snow height in winter and water residence time the rest of the year. You should at the minimum provide some references to explain why you can make such assumption.

Because these sites (IJ1 and IJ2) were located on the Pacific side of Japan (Monsoon area), the major rain (& snow) depositions were in summer. The deposition in summer (JJA) constituted 37% of the annual deposition, while that in winter (DJF) constituted 16% only. Thus, even if the actual winter aerial deposition rate (=snow deposition rate) at site IJ2 was half of that at site IJ1 due to the topographic difference, the difference between the annual aerial deposition rates (and thus annual discharge rates) at IJ1 and IJ2 would be less than 10%. Besides, the difference in water residence time have little influence on annual discharge rates under steady state condition. Because we allowed an error range of 10% in the discharge rates, we don't think this assumption is "pretty big". Further, even if the actual annual discharge rate at IJ2 deviated from the annual discharge rate used in this study due to a smaller actual deposition rate than that used in this study (50% of that used in this study, for instance), the Matm and the Datm at IJ2 would become 50% of the present estimate, so that the Matm/Datm ratio (shown in Fig. 8) would remain constant. We therefore conclude that the present assumption regarding the discharge rate has little influence on the final conclusions of this study.

> Page 7/L.22: Please provide the proportion of samples collected during the winter period vs the rest of the year.

We will make the suggested revision.

> Page 8/Section 2.4: I have several concerns regarding this section. First, regarding the extended amount of time you left the collecting bottle at the KJ site (around a

month), and considering that in summer you have temperature as high as 34C (according to section 2.1), how can you be sure that your nitrate concentrations are not biased by partial evaporation of the rain water? Oppositely, do you have a heating system to melt the snow in winter? If not, how can you be sure that you really collect 100% of wintertime precipitation? Second, you state several factors that could impact the nitrate concentration in deposition samples (incomplete dry dep collects, possible nitrification). It would be useful that you provide an associated uncertainty to the estimated atmospheric NO3-atm concentration due to these factors (and also water evaporation). > Third, you say that the deposition collector at the KJ site was installed in an open field. You must be aware that rainfall in open field is not representative of throughfall that actually reaches the soils and streams under forest canopies. For instance, Guerrieri et al. (2015) suggested that in forests with high N dep (which is the case at the KJ catchment) canopies play a significant role in modifying both NO3- concentration and isotopic composition from rainfall to throughfall. This is an important point that need to be clarified as it could impact the interpretation of your results (see below).

While the progress of partial evaporation in the collecting bottle will bias nitrate concentrations as you point out, the deposition flux will be the same. If the volume is reduced to 50% due to partial evaporation, for instance, the nitrate concentration will become 200% of the original, while the water volume will become 50% of the original, so that the deposition flux of nitrate will remain the same.

Because the atmospheric observatory at site KJ was a simple on site observatory in the forested field, all the concerns you pointed out are possible. Thus, we estimated the possible range of errors due to the bias through comparison with data determined at a nearby national atmospheric observation site (Sado-seki National Acid Rain Monitoring station), where the deposition rates were determined based on the EANET protocol (EANET, 2014); we found that all these concerns are minor. (See section 3.1 for details)

In brief, the determined deposition rates at site KJ agree (< 10% difference) with those determined at the Sado-seki National Acid Rain Monitoring station. Thus, we used the

results obtained through the observation at site KJ, allowing a moderate error range (20%) (see section 3.1).

While the differences between the $\Delta 17O$ values of nitrate in throughfall and rainfall could influence the identification of the places within the catchments where the major portions of nitrate metabolism and nitrification occurred, they had little impact on the final estimates of Catm and Matm/Datm ratio, when (1) the deposition rate of atmospheric nitrate on "the surface of catchment", in which leaf surface and crown were included, was the same with Datm used in this study within the error range ($\pm 20\%$), and (2) the average $\Delta 17O$ value of atmospheric nitrate prior to reaching "the surface of catchment", in which leaf surface and crown were included, was the same with $\Delta 17Oatm$ used in this study within the error range ($\pm 3\%$). That is to say, the nitrate in throughfall showing a different $\Delta 17O$ value than $\Delta 17Oatm$ was no longer atmospheric nitrate as per the definition of atmospheric nitrate presented in section 1.2.

> Page 10/L.14: Does it mean that in the end, all samples were analyzed, and none rejected?

Yes.

> Page 10/L.15-21: So, the values given in the manuscript are the averages of these repeated analyses?

Yes, they are.

> Page 10/L.26: Please describe what would be the highest uncertainty caused by presence of nitrite in a sample on the D17O value of nitrate (highest uncertainty would be for NO2- = 0.049 umol.L-1 and the lowest NO3- concentration you measured in your study). This would give the reader a better idea for why you regard nitrite concentrations as negligible.

We will make the suggested revision.

> Page 11/L.10-14: It might be worth illustrating this by a figure that you could place in

the SI, for readers unfamiliar with the different definitions of D17O and their discrepancies.

Many studies have been reported regarding this; in those reports you can find the luculent figures you want (Bao et al., Ann. Rev. EPSL, 2016, etc). While the difference in definition will have little influence on the results presented in this study (see section 2.6 for the details), lengthy notes and citations will be required to explain the figure to "readers unfamiliar with the definition of $\Delta$17O". Thus, we don't think the figure you requested is suitable for our paper.

> Page 12/L.6-12: A detail, but I don't think that Matm, Mre, Mtot and Datm can be classified as "fluxes", but more as "annual loads".

All the parameters (Datm, Matm, Mre, and Mtot) were certainly the "fluxes" per unit area of each catchment. While we also recognized that the term "loads" has often been used to refer to these parameters, using "loads" for those exported from the catchment (Matm, Mre, and Mtot) will be misleading to readers not so familiar with forested catchment studies. Besides, we used the same term, "fluxes" in our previous papers (Tsunogai et al., 2014) without any trouble. We have, thus, not changed terminology in this case to avoid confusing readers.

> Page 12/L.13 Add "annual" before "deposition".

We will make the suggested revision.

> Page 12/L.23: I disagree, in 2014 the flux is still higher in spring than in December. You could probably replace "December" by "winter period".

Please note that we were talking about Ftotal, not Fatm, here. The highest Ftotal of 2014 was in December (337.7 $\mu$mol m–2 day–1). This Ftotal was larger than that in June 2014 (336.0 $\mu$mol m–2 day–1). The largest Ftotal of 2013 was found in December as well (698.4 $\mu$mol m–2 day–1).

> Page 12/L.25: In 2015, you also have a June-July maximum in Fatm that is noticeable.

As opposed to the winter maximum, we could not find reproducibility in the June-July maximum. When we estimated the periodic average Fatm for two months, the June-July maximum disappeared. We don't think this maximum is worth discussing.

> Page 13/L.5: If I understood correctly, you calculated a Fatm value for every D17O value measured in the stream. Which means 15 points over two years, according to Figure 3.

Yes, you understood correctly.

> Page 13/L.5: I think this temporal resolution is way too low to accurately catch the seasonality in stream atmospheric nitrate export (and for instance snowmelt in spring). How can you make a strong statement on this subject with seven samples per year? You really need to argue hear why such low sampling resolution is enough to describe the behavior of your catchment.

Here in the section we simply wrote "We could not find significant enrichment of Fatm in spring." We don't think this is a "strong statement". Because this section presents the results of our study, we don't want to include a detailed discussion here, as per your request.

As for your comment on the number of data points (i.e., whether 15 data in total was significant), please note that the residence time of water is longer than a few months for most forested catchments in Japan with a humid temperate climate (Takimoto et al., 1994; Kabeya et al., 2007), as presented in the manuscript. That is to say, seasonal variation in the deposition rates of rain water and atmospheric nitrate in the forested catchments in Japan will be buffered by groundwater.

The almost stable $\Delta17O$ values of stream nitrate also support the fact that the rain water (and atmospheric nitrate) deposited at site KJ was buffered by groundwater in the catchment. While the total number of data points were 15 for nitrate isotopes

during the 2 years of observation at site KJ , the most nitrate-enriched sample (Dec., 2013) and the second most nitrate-depleted sample (Oct., 2014) were included in the dataset. Nevertheless, the 1 sigma variation range was less than 0.4‰ for Δ17O values of stream nitrate. The "7 data per year" (=15 data in total) was significant to determine the average Δ17O of nitrate in groundwater at the site and to characterize this catchment.

Additionally, while the stream atmospheric nitrate export was obtained from 15 data in total, the stream nitrate export, which can be estimated from nitrate concentration and flow rate, was obtained from 12 data per year. In addition to the data presented in this study, we have monthly data since 2002 (Kamisako et al., 2008; Sase et al., 2012). For instance, none of these data supported Ftotal enrichment in spring, while all supported Ftotal enrichment in winter. Because it is difficult to assume temporal changes only in Δ17O without the export flux of nitrate, our observation ("We could not find significant enrichment of Fatm in spring.") was valid.

> Page 13/L.6-7: It would be useful to the reader if you could compare here your stream nitrate concentration with other concentration measured in forested catchments outside of Japan, just like you do for the isotopic composition of nitrate later on.

We will make the suggested revision.

> Page 13/L.15: up to 1.6 mmol.L-1, or more? If it is more, then put the maximum value. As it is now, it does not make any sense.

We will make the suggested revision.

> Page 13/L.23-27: This is interpretation of the results: therefore, it needs to be removed from the Results section and moved to the Discussion section.

We will make the suggested revision.

> Page 14/L.4-18: It is more standard to present deposition fluxes in kg-N ha-1 yr-1. This would be easier to compare with other studies and to understand how elevated

deposition is on your catchments (and it is very elevated!). You may want to change your stream fluxes as well to be homogeneous on the units you use.

We are sorry, but we are not familiar with the unit kg-N ha–1. While the unit seems to be traditionally used in the studies performed in forested catchments, the unit in weight will be inconvenient to compare the results with components other than N (such as C, P, S, etc.). We have used the same unit in our previous papers (Tsunogai et al., 2010; 2011; 2014; 2018) without facing any issues. We would like to use the same unit in this paper to avoid confusing readers. We have presented the values in the unit kg-N ha–1 together with the present unit (mmol m–2 yr–1) where required.

> Page 14/L.17: Ok, so here is the error associated with your deposition estimation. Please refer to this section where appropriated earlier in your manuscript.

In the earlier sections where we talk about deposition rate estimation (such as section 2.4), we will mention that errors will be discussed in section 3.1.

> Page 14/L.21-22: Why don't you present also Fatm and Ftot for IJ2? If you don't present these results, then remove the section where you say that you extrapolated discharge data for IJ2 from IJ1 using the catchment area as converting factor.

We will add a new figure 3(e) in which Fatm and Ftotal for IJ2 are presented.

> Page 14/L.23-26: This is very far stretched. Did you conduct a statistical test to verify the decreasing trend in concentration? Also, how can you say it started in 2000 when you report three years of data covering 2012-2014?

The logic is simple. Please read carefully. Nakahara et al. (2010) found a continuous increasing trend in the annual average stream nitrate concentration, from 22 $\mu$mol L–1 in 1989 to 42 $\mu$mol L–1 in 2002. On the other hand, we found the annual average nitrate concentration to be almost stable at 24.4 $\mu$mol L–1 from 2012 to 2014. To connect these different trends in the same stream (IJ1) continuously and smoothly, we needed to assume the turning point from increasing to decreasing trend. Thus, we wrote "the

trend in stream nitrate concentration has changed from increasing to decreasing". We did not insist that the annual mean nitrate concentration at IJ1 was decreasing during our observation period (2012 to 2014).

Further, the turning point must have been between 2003 and 2011. If we assume the turning point to be immediately before our observation period (2010 or 2011), the trend would be discontinuous. Besides, it would be difficult to explain why the annual average nitrate concentration was almost stable from 2012 to 2014. Thus, we estimated that the turning point was in the 2000s, between 2003 and 2009, and wrote "probably since the 2000s".

> Page 16/L.10: Why not giving the corresponding proportions of nitrate for each source using D17O to calculate the percentages.

These are presented in section 4.3 as Matm/Mtotal.

> Page 16/L.17-20: This sentence doesn't make sense. How can D17O of soil nitrate reflect the original value of nitrate in groundwater?

In L17-18 we write "the $\Delta$17O values of soil nitrate REPRESENTED the original $\Delta$17O values of nitrate in the groundwater". Because all stream nitrate data were plotted at the central part of the region produced by soil nitrate in Fig. 4, this must be a reasonable explanation for the relation between stream nitrate (=groundwater nitrate) and soil nitrate.

> Page 16/L.17-20: I think that soil nitrate reflects atmospheric nitrate D17O value, that is buffered by nitrification in soils (as shown by the seasonal variation in figure 3). Soil nitrate isotopic composition is not related to groundwater nitrate isotopic composition.

We feel that your interpretation does not apply to the data presented. As clearly presented in Figs. 3(c) and 5, nitrification must be active in summer soil and inactive in winter soil. Without assuming nitrate in groundwater (i.e., as "buffer" we proposed), it is impossible to explain the much smaller temporal $\Delta$17O variation in stream nitrate than

that in soil nitrate.

The soil nitrate in winter, for instance, always showed higher $\Delta17O$ values than that of stream nitrate, irrespective of the location of sampling point and depths of catchment. That is to say, the proportion of nitrate produced through nitrification (i.e., the "buffer" you proposed) within soil nitrate in winter was smaller than that in stream nitrate in winter. Of course, the $\Delta17O$ values of nitrate in rain/snow were higher than those in stream nitrate. Therefore, it is impossible to explain the lower $\Delta17O$ values of stream nitrate in winter with your interpretation, where you did not assume the contribution of the nitrate in groundwater (i.e., as "buffer" we proposed).

> Page 17/L.13: You need to indicate how you come up with this value. If you used the following equation: $\_18O\text{-}NO\text{-}3 = 1/3 (\_18O\text{-}O2) + 2/3 (\_18O\text{-}H2O)$, then you must also comment on the limitations of its utilization (see Rose et al. 2015a and Snider et al. 2010).

We used the relation presented in Buchwald et al. (2012). This was different from the equation you suggest. We will present the equation in the manuscript, in response to your request.

> Page 17/L.13: You should also try to plot d18O vs d15N and see how they correlate to confirm, or not, the absence of biological processes in your catchment. > Page 18/L.9: Here you should provide a scatterplot of d18O vs d15N to discuss the presence, or not, of any correlation between these isotopic values throughout the year. And even if you don't find any correlation, it does not mean that no assimilation is taking place (it would be really surprising to have no assimilation anywhere in the catchment) but rather than the recharge of new NO3-re is overprinting isotopic fractionation by assimilation (Granger and Wankel, 2016).

First of all, we did not intend to say that biological processes were absent in the catchment. What we wrote here was that "partial metabolism was MINOR (= not so active) in the catchment". If "minor" is a misleading word, we will change it to a more appropriate

expression.

We used a $\delta15N$ ($\delta15Nre$) vs. $\delta18Ore$ plot for stream nitrate eluted from a forested catchment in our previous paper (Tsunogai et al., 2014) to investigate the source and behaviour of remineralized nitrate. We also estimated $\delta15N$ and $\delta18Ore$ in this study, but we could not find any significant correlation between $\delta15N$ and $\delta18Ore$ in stream nitrate at site KJ ($R2 = 0.06$). However, we found significant variation in both $\delta15N$ and $\delta18Ore$, around 5‰ in stream nitrate and more than 10‰ in soil nitrate.

Unlike in our previous study where nitrate metabolism simply controlled $\delta15N$ and $\delta18Ore$, the controlling factor for $\delta15N$ ($\delta15Nre$) of nitrate seems to be complicated at site KJ. To interpret the results accurately, however, many more pages and data would be needed. We are working towards this in a future article.

> Page 17/L.17-19: This statement needs to be amended or removed. This is hardly new results, as the relationship between D17O and d18O have been used for almost a decade to understand biological processes in catchments (see previous work by Tsunogai et al.).

While we first estimated the average $\delta18Ore$ in the catchment using the relationship between $\Delta17O$ and $\delta18O$ (Tsunogai et al., 2010), this work has often been ignored in many of the subsequent works (Dejwakh et al., 2012; Liu et al., 2013; Riha et al., 2014; etc.), which estimated $\delta18Ore$ (or $\delta18Oterr$ or $\delta18Obio$ in some cases) without citing our study. That is to say, Tsunogai et al. (2010) is still "new" (i.e., still not known) by many people studying $\Delta17O$ of nitrate. We would like to emphasize the usability of this methodology again here, citing our previous work.

> Page 17/L.21-23: Is that an assumption or something you know for a fact? Please provide the data to justify that rain is responsible for Ftotal increase for winter (either the precipitation chart, or some data indicating that there is more precipitation in winter than I summer).

That was a fact. You can find the evidence in the high flow rates of the stream in winter. We will make the suggested revision, by adding deposition rate data in section 2.1.

> Page 18/L.6: Add reference to Michalski et al. 2004 after the value of 0.

We will make the suggested revision.

> Page 18/L.18-19: Be more precise: you are talking about concentrations here.

Yes, we are.

> To say "stream nitrate shows a normal correlation with soil nitrate" doesn't make any sense.

We will revise this to "stream nitrate concentration shows a normal correlation with soil nitrate concentration".

> Also, I am a bit dubious of the strength of your correlation given the low number of samples presented in Figure 7 (n=11).

This was the reason we calculated the p-value.

> Page 18/L.20-25: Alternatively, the slightly higher D17O values in winter/spring comparted to the rest of the year are due to freeze/thaw events leading to partial snowmelt, that is not well captured by your very coarse sampling resolution. How can you exclude that hypothesis, especially as you said that the KJ catchment is covered by snow from December to March, exactly when the Fatm is higher? This needs at least to be discussed. Didn't you measure water isotopes as well? Can't you tell from these measurements if the water comes from groundwater or from snowmelt (Hall et al., 2016; Liu et al., 2004)? That would be a very strong addition to your reasoning.

We can find high $\Delta17O$ at the end of July (+1.5‰ in 2013 and +1.6‰ in 2014). The $\Delta17O$ value in 2013 was the 2nd highest $\Delta17O$ and the $\Delta17O$ value in 2014 was the 3rd highest $\Delta17O$ among the $\Delta17O$ data obtained each year studied (n=7, respectively). It was impossible to explain the higher $\Delta17O$ found at the end of July in both

years by the "freeze/thaw events", because all the snow disappeared by the end of April every year.

Further, we determined both $\delta 18O$ and $\delta 2H$ values of the samples of stream water. Both values in the stream water showed little seasonal variation (-9.1±0.3 ‰ and -48.6±3.0‰ respectively). The d-excess (=$\delta 2H$ –8*$\delta 18O$) in the stream water showed little seasonal variation as well (+24.2±1.9‰. Because d-excess in rain (&snow) water in these regions (Japan sea side of eastern Japan) shows large seasonal variation every year (around +30‰ in winter and around +10‰ in summer; Tanoue et al., 2013), the water isotopes also supported our hypothesis, while the contribution of water from the "freeze/thaw events" was minor in the stream water.

We will make the suggested revision in 4.1 adding the reference (Tanoue et al., 2013). Additionally, we will present the data of water isotopes in supplement.

Tanoue, M., K. Ichiyanagi, and J. Shimada (2013) Seasonal variation and spatial distribution of stable isotopes in precipitation over Japan, J. Jpn. Assoc. Hydrol. Sci., 43(3), 73-91 (in Japanese with English abstract).

> Page 19/L.1-15: It would be nice here that you refer to a table where you list the annual mean values for Catm, Ctot, Fatm, Ftot, and Matm for each watershed. It is hard to keep up with all you say because we are always looking for the values elsewhere in the manuscript.

We will make the suggested revision.

> Page 19/L.16: There is no Equation (9) in your manuscript

We are very sorry, but the numbering of the equations seems to have been removed while we were arranging the format for Biogeosciences Discuss. The equation (9) was the last equation presented in section 2.7. We will make these revisions.

> Page 19/L.17: You need to detail how you calculate your incertitude either in the Methods or in the SI: you stipulated earlier in the manuscript that you would assume a

20% error on the Datm at the KJ catchment (see Page 14, L.17). Can that result in the 2.6% error on the Matm/Datm ratio that you present here? Did you perform a formal error propagation calculation? Please expand more on this aspect.

Twenty percent of the Matm/Datm ratio showing 9.4% (=0.094) corresponds to 1.9% (=0.019). Because Matm also includes an error, the final error of the Matm/Datm ratio was 2.6%. These are simple calculations of error propagations that were beyond the scope of this study.

> Page 19/L.17: Also, I would be very curious on how you obtained your percentages. If I divide the Matm (=8.8, 5.7, and 2.2 for KJ, IJ1 and IJ2 respectively) by Datm (=45.6, 49.2, and 48.3 for KJ, IJ1 and IJ2 respectively), I obtain 19.3, 11.6 and 4.8%. Not at all what you calculated. Please explain

We are very sorry, but the Matm values presented here were total Matm values during the observation (ca. 2 years) and the annual Matm value was approximately 50% of the present. We will present the annual values for Mtotal and Matm.

> Page 20/L.7-8: I don't understand what in Figure 9 can lead to such conclusion: I am pretty sure that contrary to what you say, different nitrification rates in soils will lead to different NO3-re leaching fluxes to stream/groundwater, and thus impact the D17O value of nitrate in stream (by dilution). This would impact Matm, and therefore the Matm/Datm ratio. So please clarify what you meant here.

Your statement "This would impact Matm" does not apply. While different nitrification rates in soils lead to different NO3–re leaching fluxes to stream/groundwater, and thus, impact the $\Delta17O$ value of the nitrate in the streams (by dilution), different nitrification rates cannot impact Matm, because Matm is determined by the processes of (1) deposition rate of atmospheric nitrate (Datm), and (2) metabolic rate of nitrate in the forested catchment. Under the same removal rate constants for nitrate metabolism, and the same residence time of water in the catchment, Matm values are stable irrespective of the changes in the nitrification rates.

> Page 21/L.10-13: That is my point. Rose et al 2015 already showed the relationship between Matm and N saturation? So, what is new here? What does your ratio bring more than just the use of Matm? Also, please try to plot the same correlation with Matm instead of Catm, and report the correlation strength. > Figure 8: Please add a third panel to show Matm vs Ctot.

The metabolic rates of nitrate in forest soils that determine the removal ratio of atmospheric nitrate from the total atmospheric nitrate deposited in a catchment ((Datm – Matm)/Datm ratio; Tsunogai et al., 2014), not the Matm. Because Matm is a function of Datm, using Matm/Datm ratio as the index, instead of Matm, is essential in principle.

In response to your strong request, we will add a new figure 8(c) in which the relation between the nitrate concentrations and Matm values is plotted. The correlation coefficient (R2= 0.63) was lower than those of the average nitrate concentration vs. Matm/Datm ratio (R2= 0.92) and average nitrate concentration vs. Matm/Mtotal ratio (R2= 0.80).

We would like to thank you for the helpful comments and suggestions. We trust that our responses to your comments and questions are satisfactory.

Sincerely, Urumu Tsunogai

Cc: Fumiko Nakagawa, Yusuke Obata, Kenta Ando, Naoyuki Yamashita, Tatsuyoshi Saito, Shigeki Uchiyama, Masayuki Morohashi, Hiroyuki Sase

Please also note the supplement to this comment:
https://www.biogeosciences-discuss.net/bg-2018-258/bg-2018-258-AC1-supplement.pdf

---

## Author Comment (AC2) · 16 Sep 2018

Reply to your comment (Referee #2).

Thank you very much for your valuable comments on our manuscript. We have responded to each of your comments and questions.

> First, the watersheds (KJ, IJ1 and IJ2) have extremely different size (3.84, 298, 108 ha respectively) and there are no information concerning their geology and soil depth/quality which are important parameters to interpret stream discharge and nutrient flow. > Experimental section: KJ, IJ1 and IJ2 have very different size. Here (and

probably in the discussion) you should somehow reassure the reader that it is not a source of bias in your interpretation.

In the same Niigata Prefecture in which KJ is located, Koshikawa et al. (2011) determined stream chemistry from streams (n=62) having various catchment areas, ranging from 0.007 km2 (=0.7 ha) to 18 km2 (=1800 ha). They performed primary component analysis (PCA) of the various factors possibly related to the stream chemistry including nitrate concentration, and, they could not find any significant relation between stream nitrate concentration and catchment area. Thus, the differences in the catchment area (from 0.7 to 1,800 ha) had little impact on the stream nitrate concentration. As a result, we targeted site KJ (3.94 ha) for analysis. We would like to present this information in section 2.1 along with this reference.

Koshikawa, M., M. Watanabe, T. Takamatsu, S. Hayashi, S. Nohara, and K. Satake (2011) Relationships between stream water chemistry and watershed geology and topography in the Miomote River System, Niigata, Japan. Japanese Journal of Limnology, 72, 71-80 (in Japanese with English abstract).

The geology and soil quality of the sites has been previously studied. In brief, the bedrock consists of granodiorite, and brown forest soils have developed at site KJ (Kamisako et al. 2008; Sase et al. 2008). The bedrock consists of the chert (90%) and mudstone (10%) of Middle Jurassic to Early Cretaceous age at sites IJ1 and IJ2 and the dominant soil type was brown forest soils (Nakahara et al., 2010). We will add information on geology and soil quality in the revised MS.

> Second, stream discharge is estimated with a different method for each stream (section 2.2). > Section 2.2: Discharge is estimated with different methods for each stream. This is a source of variation for your results and you should acknowledge that somewhere in the discussion to convince the reader you took this issue into account in your interpretation.

Each method to estimate stream discharge was the same as those used in previous

studies (site KJ: Kamisako et al. 2008; IJ1 and IJ2: Nakahara et al. 2010). Both methods are commonly used well established methods to measure stream discharges (U.S. Department of the Interior Bureau of Reclamation, 1997). Additionally, Kamisako et al. (2008) confirmed that the water balance was justified at site KJ, in which the evapotranspiration loss was estimated based on Thornthwaite's (1948) method. We concluded that the discharge rate at site KJ was accurate (within the error range of 10%) and used in the discussions in this MS. In IJ1 and IJ2, there were no prior studies on water balance. The estimated annual discharge via the stream (2,057 mm on average during 2013–14) corresponds to 62% of the annual deposition in IJ1 (3,310 mm on average during 2013–14). Because the evapotranspiration loss from forested catchments in Japan was estimated to be 30 to 40% of deposition under the annual deposition rate of 3000 mm (Ogawa et al., 2003), we concluded that the estimated annual discharge via the stream was highly reliable in the sites as well, within the error range of 10%. We will add this information concerning the water balance in the revised MS, together with references.

Ogawa, S. (2003) Chapter 3, Forests and water resources, In: Hydrological Cycle and Local Metabolic System of Water (eds. N. Tambo and T. Maruyama), Gihodo, pp. 45-71 (in Japanese).

U. S. Department of the Interior Bureau of Reclamation (1997) Water Measurement Manual. US Government Printing Office: Washington, DC.

> Third, atmospheric nitrate deposition rates were also obtained with different approaches in the three watersheds (section 2.4). > There is the same risk of discrepancy with the way atmospheric nitrate deposition rates were obtained (section 2.4).

While the deposition rates of atmospheric nitrate at sites IJ1 and IJ2 were determined at a National Acid Rain Monitoring station based on the EANET protocol (EANET, 2014), those at site KJ were determined by a simple on site observation in the forested field, so that the deposition rates could be biased to the same extent, as presented in

sections 2.4 and 3.1.

As a result, we compared the deposition rate at site KJ with those determined at the nearby national atmospheric observation site (Sado-seki National Acid Rain Monitoring station), where the deposition rates were determined based on the EANET protocol (the same protocol with IJ1/IJ2 sites), and found that the deposition rates at site KJ agreed (< 10% difference) with those determined at the Sado-seki National Acid Rain Monitoring station. Thus, we concluded that the methodological differences had little impact on the final results and we used the results obtained through the observation at site KJ, allowing a moderate error range (10%). (See section 3.1 for details)

> Finally one can wonder how much data variation is carried by each of these points and how this influences the results and their interpretation. My guess is that this should be at least acknowledged, analysed and discussed thoroughly to convince the reader.

For the isotopes ($\delta15N$, $\delta18O$, and $\Delta17O$), we repeated the analyses at least three times and obtained the results presented in the manuscript as the average (P10/L15). While the errors associated with the $\Delta17O$ values directly influences the errors associated with the Catm/Ctotal ratios and Matm/Mtotal ratios, their influences on the values of Matm, and Matm/Datm ratios were minor. Rather, the errors associated with the flow rates (V in Eqs. (3) and (4)) and the deposition rates (Datm) had much larger impact on the Matm and Matm/Datm ratios.

We will make the suggested revision in section 5, where we have presented the major factors influencing the errors associated with the Matm and Matm/Datm ratio.

> Abstract: L21: I suggest you give the annual fluxes of nitrate instead of concentrations.

We will make the revision suggested.

> Abstract: L23-25: It is misleading to compare, in the same sentence, 17O-excess in KJ soil water with those of KJ, IJ1 and IJ2 stream water. Add "in KJ" after "+0.1‰

to+5.7‰$''.

We will make the revision suggested.

> Abstract: L25: "was groundwater nitrate".

We will make the revision suggested.

> Abstract: L25: I would also remove the end of the sentence "which buffered the seasonal. . ." as it is confusing.

We would like to revise here.

> Introduction: P2-L5 (and throughout the paper): Do not use the word "enrichment" when you simply mean "increase", especially in a paper dealing with isotopes for which "enrichment" is usually used as in P4-L7.

We will make the revision suggested.

> P2-L25: There are other processes that should be mentioned: DNRA for instance.

We will change No. 4 to "(4) the removal of nitrate through dissimilatory reduction by microbes".

> P3-L2-3: Remove the end of the sentence ", including [. . .] catchments.", it does not bring new information.

We will revise here to bring new information.

> P3-L4: composition. . .has been (no plural)

We will make the revision suggested.

> P3-L8: "has survived"

We will make the revision suggested.

> P3-L13: remove "nitrate including"

We will make the revision suggested.

> P3-L21-22: this sentence is useless. I miss some clear hypotheses at the end of the introduction.

We will revise this sentence, and add a sentence here to present the hypothesis of this study more clearly.

> Section 1.2 is Materials and Method.

In section 1.2, we presented the background of this study (regarding the $\Delta$17O tracer of nitrate), because this tracer is not very familiar for those studying Biogeosciences.

> P4-L6: atmospheric nitrate is not only produced by photochemical reactions.

We will add "most of" prior to "which is produced via photochemical reactions . . .".

> P4-L13: specify this is Equation 1 > P4-L24: Specify this is Equation 2

We are very sorry, but the numbering to the equations seems to have been removed while we were arranging formatting the paper for Biogeosciences Discuss. We will make this revision.

> Here you use Catm and Ctotal for the first time. In the rest of the manuscript they represent either mean daily or mean annual concentrations which is confusing. Please use different codes for daily and annual.

We now use Catm and Ctotal only for daily mean concentrations (i.e., concentrations in each sample) and use different characters (atm and total) for annual mean concentrations in the revised MS.

> Experimental section: P5-L15: Remove "continuous" or replace by "weekly" as samples were collected once a week.

We will make the revision suggested.

> P5-L25: "2500 mm" seems a lot as the average precipitation in Shibata is 1263 mm.

1. The average precipitation in Shibata is estimated to be 1920 mm (https://en.climate-data.org/region/2441/#example2). Your precipitation data (1263 mm) may be inaccurate.

2. The annual precipitation determined at the AMeDAS (Automated Meteorological Data Acquisition System) observatory nearby (Nakajyo; 38°3.6'N; 139°24.5'E) is 2110 mm (2012), 2746 mm (2013), and 2506 mm (2014), so that the precipitation in Nakajyo supported the "2500 mm" value. You can find the data of AMeDAS on the web site of Japan Meteological Agency (http://www.jma.go.jp/jma/indexe.html).

3.ãĂĂWhile Shibata is located on a flatland, Nakajyo is located at the eastern foot of the Kushigata mountain range so that the precipitation at Nakajyo is higher than that at Shibata. Because site KJ is located at the eastern edge of the Kushigata mountain range, we conclude that the precipitation at Nakajyo represents the precipitation in KJ.

> P6-L9: "3300 mm" seems a lot. Actually if I do the maths with your figures I get 1500 mm for 10 months/yr.

1. The annual precipitations determined at the AMeDAS observatory nearby (Tarumi; 35°38.3'N; 136°36.2'E) was 3414 mm (2012), 3021 mm (2013), and 3497 mm (2014), so that the precipitation data in Tarumi support the "3300 mm" value.

2. Figure 6 (a) shows the flow rate at IJ1. When we integrate the flow rates determined in this study (Figure 6 (a)), the annual discharge via the stream corresponds to 2057 mm for the catchment (298 ha) on average from 2013 to 2014, which almost corresponds to your rough estimate (1500 mm). To compare annual discharge with precipitation, however, we should consider the evapotranspiration loss from the catchment as well. Because the evapotranspiration loss from forested catchments in Japan was estimated to be 30 to 40% of deposition under the annual deposition rate of 3000 mm (Ogawa et al., 2003), we think the annual precipitation of 3300 mm is accurate for IJ1.

Ogawa, S. (2003) Chapter 3, Forests and water resources, In: Hydrological Cycle and Local Metabolic System of Water (eds. Norihito Tambo and T. Maruyama), Gihodo, pp. 45-71 (in Japanese).

> P6-L23: I am not sure Clethra sp. and Ilex sp. can be considered as trees.

While there is no precise botanical definition for tree, we confirmed that both Clethra harbinervis and Ilex pedunculosa were classified as trees in a Pictorial Book of Flora in Japan.

> P7-L1: You do not use water T, pH, alkalinity and EC in the paper. Please remove. In fact, maybe it could bring new insights in your data set and interpretation.

While these data were routinely obtained to check the reliability of sampling, we do not present the data as you pointed out, because we could not find anything new in the results than those reported in previous studies (Kamisako et al., 2008; Nakahara et al., 2010). We are presenting this data in a supplement.

> Section 2.3: what SLS and SMS stand for? You need to explain why you decided to sample next to the stream and 20m upland, and why at 20cm and 60cm depth.

The depths were chosen as the length of the porous cup soil solution sampler (20 cm and 60 cm). We will add this in the revised MS, in response to your request.

The three soil water sampling stations were chosen on the transect from the sampling station for atmospheric deposition toward the direction perpendicular to the slope (and thus almost perpendicular to the stream). In addition to the lower station (SLS) and middle station (SMS) on the transect, there was one station (SUS) at the upper end of the line. The SUS station, was hard to access and this limited the number of samples collected there (most of them were taken only in summer). Further, the volumes of solution for isotope analysis were also limited. As a result, we did not analyse isotopes for the samples and thus, we did not present the results.

Because we could not find any significant differences in the isotopes irrespective of the

sampling depths and stations (Fig. 3), we concluded that the data on isotopes obtained at SLS and SMS represented that of soil solutions at the site. We will emphasize this in the revised MS.

> P7-L22: "between December and March"

We will make the revision suggested.

> Section 2.5: I am surprise samples were not acidified prior storage.

The oxygen isotope exchange reaction between NO3– and H2O becomes much faster under low pH conditions (Böhlke et al., Rapid Commun. Mass Spectrom., 17, 1835-46, 2003; Kaneko and Poulson, Geochim. Cosmochim. Acta, 118, 148-156, 2013). We cannot store samples for oxygen isotope analyses of nitrate under acidified conditions.

> P9-L15: remove ", the procedure [. . .]. Approximately"

We will make the revision suggested.

> P10-L22: You mention NO2 as a possible source of variation, what about NH4?

Nitrogen compounds that can be converted to N2O on addition of azide (N3H) can interfere with the results. To convert NH4+ to N2O, we must add an oxidizing agent (such as NO2–), while azide is a reducing agent. Then, NH4+ cannot interfere with the results.

> Section 2.7: I suggest you give the units for each Equation.

Because these are general equations and no actual value was presented, there are no constraints on the units.

> P12-L1: replace "obtain" by "estimate"

We will make the suggested revision.

> Results: P12-L16-21: You can remove this paragraph which does not bring new information compared to the figures legends.

We will make the suggested revision.

> P12-L25: Could you add ranges or values to Ftotal and Fatm. Same thing for IJ1 and IJ2.

We will make the suggested revision.

> P14-L20-22: You can remove this sentence which does not bring new information compared to the figures legends.

We will make the suggested revision.

> Discussion: Specify to which category (catchment groundwater, through flow) belongs the water sampled in SLS and SMS.

We will make the suggested revision.

> P16-L25: Add to Fig 4 legend that the hypothetical mixing line is reported.

We will make the suggested revision.

> P17-L1: replace "buffered by" by "diluted in"

While we estimated that the total quantity of groundwater nitrate was much more than that of soil nitrate, the concentration of soil nitrate was not always higher than that of groundwater nitrate. As a result, we cannot agree with you on the use of the word "dilute" here for nitrate. We will revise here, in response to your request, but different way than what you suggest.

> P18-L22: "This delay time reflects the magnitude and flow of the nitrate. . ."

We will make the suggested revision.

> P19-L23-24: A figure showing the relationship between Matm and stream NO3 conc. would be welcome.

We will add this as a new figure 8(c).

**BGD**

> P20-L3: replace "N" by "NO3" > P20-L4: replace "nitrogen" by "nitrate".

While we can use NO3– (nitrate) instead of N (nitrogen) in this text, what the original literature (Peterjohn et al., 1996) used was N (nitrogen). We would like to use the same here.

> P20-L8: Fig. 9 does not show that "Matm/Datm ratios are stable during the progress of nitrification in forested soils"

Please see the locations of the arrows (=flows) representing nitrification. All the arrows related to the Matm/Datm ratios are shown in red/pink, while those related to nitrification are shown in brown/yellow. The colour differences represented the Matm/Datm ratios were determined independent of nitrification. We will add this explanation in the text and the caption to figure 9.

> P20-L17-22: this paragraph is not clear enough. In particular the last sentence compare"stream nitrate enrichment due to nitrogen saturation" with "stream nitrate enrichment due to artificial processes", which I do not quite understand. Artificial processes (e.g. fertilizers, leguminous fields. . .) are responsible for N saturation. Please clarify.

The "artificial processes" we presented here correspond to the direct contamination processes of nitrate. Increases in stream nitrate concentrations due to secondary changes in the N cycles within forested catchments were not included. We will revise this paragraph to reconfirm this.

> P21-L24: ". . .by 6 and 20 times respectively in accordance. . ."

We will make the suggested revision.

> Fig.2: Underneath RW1 and RW3 there are distances (120m and 40m respectively), are they elevation a.s.l.?

Yes, they are. We will clarify this in the caption of revised MS.

> Fig1 and Fig.2: Please use the same kind of map for both figures.

The base altitude maps (colour altitude maps) are produced by using the same software. While the number of colours used in the altitude map of Fig. 1 was much lower than that for Fig. 3, this is because the variation range in altitudes was much smaller in the Fig. 1 area than that in the Fig. 3 area. In addition to the base altitude map, however, we added contours in Fig. 1 because it was difficult to understand the differences in altitudes in the Fig. 1 area without the contours. We will use the same figures (with contours only in Fig. 1) to clarify high and low areas in the catchments. Also, we will use similar formats for scale bars etc. in the figures, in response to your request.

> Fig.4: Please specify what the line stands for?

We will revise this figure.

We would like to thank you for the helpful comments and suggestions. We trust that our responses to your comments and questions are satisfactory.

Sincerely, Urumu Tsunogai

Cc: Fumiko Nakagawa, Yusuke Obata, Kenta Ando, Naoyuki Yamashita, Tatsuyoshi Saito, Shigeki Uchiyama, Masayuki Morohashi, Hiroyuki Sase

Please also note the supplement to this comment:
https://www.biogeosciences-discuss.net/bg-2018-258/bg-2018-258-AC2-supplement.pdf

―――――――――――――――――――――

---

## Author Response (AR1)

Oct. 11, 2018

Dr. Sébastien Fontaine
Editor of Biogeosciences

Title: Export flux of unprocessed atmospheric nitrate from temperate forested catchments: A possible new index for nitrogen saturation
Authors: Nakagawa, F., Tsunogai, U., Obata, Y., Ando, K., Yamashita, N., Saito, T., Uchiyama, S., Morohashi, M., Sase, H.,
MS No.: bg-2018-251

Dear Dr. Fontaine:

Thank you very much for handling our manuscript. We would like to thank the referees as well for the constructive comments on our manuscript. We have carefully studied the comments and revised the manuscript accordingly. All the revisions have been listed in this letter, together with the reasons. Besides, we also uploaded the revised manuscript in MS Word, in which all the revisions from BGD version were recorded.

Major revisions from BGD are as follows:

1) As presented in our reply to the anonymous referee #1, the $M_{atm}/D_{atm}$ ratio, the directly exported atmospheric nitrate flux relative to whole deposition flux of atmospheric nitrate in a catchment area, was used in our previous study as an index to evaluate the biological metabolic rate of nitrate in forest soils in each catchment (Tsunogai et al., 2014). We emphasized this in sections 1.1 (P3/L16 in BGD, P3/L23-26 in the revised MS) and 3.5 (P19/L20 in BGD, P23/L12-27 in the revised MS) of the revised MS.

2) As presented in our reply to the anonymous referee #1, there have previously been many ecological and biogeochemical studies on the high elution rates of nitrate in the sites (site KJ: Kamisako et al., 2008, Sase et al., 2008; 2012, IJ1 site: Yamada et al., 2007; Nakahara et al., 2010). In the revised MS, we emphasized that there have previously been many studies on the sites (P6/L4 and P7/L3 in BGD, P7/L8-10 and P8/L10-11 in the revised MS) and our conclusion regarding the nitrogen saturation at the studied sites agrees well with the results obtained in these previous studies (P20/L16 in BGD, P24/L18-27 in the revised MS). Additionally, we added a reference to present that the deposition rate of atmospheric nitrogen in site KJ exceeds the threshold for nitrogen saturation proposed in the reference.

Aber, J. D., Goodale, C. L., Ollinger, S. V., Smith, M.-L., Magill, A. H., Martin, M. E., Hallett, R. A., and Stoddard, J. L.: Is nitrogen deposition altering the nitrogen status of northeastern forests?, Bioscience, 53, 375-389, 2003.

3) We used either "high concentration (of nitrate)" or "elevated concentration (of nitrate)" instead of "(nitrate) enrichment" throughout this manuscript, in response to the referees' comments.

4) In response to the referee #2's request, we added sentences in section 2.1 (P6/L3 in BGD, P7/L2-8 in the revised MS) to explain that the differences in the catchment area had little impact on the stream nitrate concentration, along with a reference (Koshikawa et al., 2011). Besides, we added information on geology and soil quality of the catchments in the revised MS (P5/L21 and P6/L15 in BGD, P6/L15 and P7/L20-22 in the revised MS).

Koshikawa, M., M. Watanabe, T. Takamatsu, S. Hayashi, S. Nohara, and K. Satake (2011) Relationships between stream water chemistry and watershed geology and topography in the Miomote River System, Niigata, Japan. *Japanese Journal of Limnology*, 72, 71-80 (in Japanese with English abstract).

5) We mixed the "Results" section (sections 3.1 and 3.2) and the "Discussion" section (sections 4.1, 4.2, and 4.3) in BGD together as "Results and Discussion" section in the revised MS, in response to the referee #1's comment on P13/L23-27 in BGD. Through the revisions, we replaced the section numbers 4.1, 4.2, 4.3, and 5 in BGD by 3.3, 3.4, 3.5, and 4, respectively.

6) In response to the referee #1's concerns on uncertainties in the stream discharge rates, we added paragraphs in sections 3.1 and 3.2 (P14/L8-18 and P16/L14-P17/L4 in the revised MS) to discuss the water balance in the studied sites and to justify the estimated annual discharge rates via stream (within the error range of 10%). We also added a reference to discuss the water balance.

Ogawa, S. (2003) Chapter 3, Forests and water resources, In: Hydrological Cycle and Local Metabolic System of Water (eds. N. Tambo and T. Maruyama), Gihodo, pp. 45-71 (in Japanese).

7) In response to the referee #1's comment on section 4.3 (P18/L20-25 in BGD), we added a paragraph in section 4.1 of BGD (section 3.3, P19/L14-25, in the revised MS) to emphasize that the water isotopes also supported our hypothesis that the major source of stream water was groundwater in site KJ. We added references as well to explain and to interpret the data on water isotopes (Dansgaard, 1964; Tanoue et al., 2013). Additionally, we presented the data of water isotopes in supplement (Fig. S1).

Dansgaard, W.: Stable isotopes in precipitation, Tellus, 16, 436-468, 1964.

Tanoue, M., K. Ichiyanagi, and J. Shimada (2013) Seasonal variation and spatial distribution of stable isotopes in precipitation over Japan, J. Jpn. Assoc. Hydrol. Sci., 43(3), 73-91 (in Japanese with English abstract).

8) In response to the referee #1's request on Page 19/L1-15 in BGD, we added a new table (Table 1 in the revised MS), in which we listed the annual mean values of $C_{total}$, $C_{atm}$, $M_{total}$, $M_{atm}$, $M_{atm}/M_{total}$, and $D_{atm}$ in each catchment.

9) In response to the referee #1's request on both P21/L10-13 and Figure 8 in BGD, as well as referee #2's comment on P19/L23-24 in BGD, we have added a new figure 8(c) in which the relation between the average nitrate concentrations ($[C_{total}]_{avg}$) and $M_{atm}$ is plotted. Besides, we added a paragraph in the revised MS (P22/L2 in BGD, P26/L16-23 in the revised MS) to explain the correlation coefficient between average nitrate concentration and $M_{atm}$ ($R^2 = 0.63$) was poorer than those between the average nitrate concentration and $M_{atm}/D_{atm}$ ratio ($R^2 = 0.92$).

10) As suggested by referee #2, we made the revision in section 5 in BGD (section 4, P27/L17-21, in the revised MS) to describe the major factors influencing the errors associated with the $M_{atm}/D_{atm}$ ratio.

11) We had used "$C_{total}$" (or "$C_{atm}$") not only for concentrations of nitrate (or atmospheric nitrate) in each sample but also for annual average concentration of nitrate (or atmospheric nitrate) in each stream in BGD. In response to the referee #2's comment on this, we used $[C_{total}]_{avg}$ (or $[C_{atm}]_{avg}$) for annual average concentration of nitrate (or atmospheric nitrate) in this study.

12) The English of the manuscript was thoroughly edited by Editage English editing service (http:// www.editage.jp/) again prior to submit revised manuscript, in response to the comments by the referee #1.

Minor revisions from BGD are as follows:

(P1/L21 in BGD, P1/L21-22 in the revised MS) We added the annual flux of nitrate in each site, as suggested by referee #2.

(P1/L23-25 in BGD, P1/L25 in the revised MS) We added "in KJ" after "+0.1‰ to +5.7‰", as suggested by referee #2.

(P1/L25 in BGD, P2/L1 in the revised MS) We replaced "was nitrate in groundwater" by "was groundwater nitrate", as suggested by referee #2.

(P1/L25-27 in BGD, P2/L2-3 in the revised MS) We revised this sentence in response to the referee #2's comment.

(P2/L11 in BGD, P2/L14 in the revised MS) We replaced "representative" by "most important", as suggested by referee #1.

(P2/L13 in BGD) We removed "receiving" as suggested by referee #1.

(P2/L15-16 in BGD) We removed "probably" as suggested by referee #1.

(P2/L25 in BGD, P3/L4-5 in the revised MS) We changed No. 4 to "(4) the removal of nitrate through dissimilatory reduction by microbes", as suggested by referee #2.

(P3/L2-3 in BGD, P3/L6-8 in the revised MS) We revised this sentence to emphasize that the processes responsible for the elevated nitrate concentrations in streams eluted from nitrogen-saturated forested catchments were not clarified as yet, in response to referee #2's comment.

(P3/L4 in BGD, P3/L9 in the revised MS) We have fixed typo in this sentence, as suggested by referee #2.

(P3/L5 in BGD, P3/L11 in the revised MS) We newly cited Kendall et al. (2007) here, as suggested by referee #1.

Kendall, C., E. M. Elliott, and S. D. Wankel (2007) Tracing anthropogenic inputs of nitrogen to ecosystems, in Stable Isotopes in Ecology and Environmental Science, 2nd edition, edited by R. H. Michener and K. Lajtha, pp. 375-449, Blackwell Publishing.

(P3/L8 in BGD, P3/L13 in the revised MS) We have fixed typo in this sentence, as suggested by referee #2.

(P3/L8-9 in BGD, P3/L13-14 in the revised MS) We have revised the definition of unprocessed atmospheric nitrate to "nitrate supplied via atmospheric deposition that has not been involved in the N cycle through the biological processing of nitrate, such as …", as suggested by referee #1. The reason for this revision had been presented in our reply to the anonymous referee #1.

(P3/L12 in BGD, P3/L18-21 in the revised MS) As presented in our reply to the anonymous referee #1, what we wanted to emphasize here was that we can quantify unprocessed atmospheric nitrate (we can determine the absolute concentration of unprocessed atmospheric nitrate) from both $\Delta^{17}O$ value and concentration of stream nitrate. So as not to mislead readers, we have revised this sentence.

(P3/L13 in BGD) We have removed "nitrate including", as suggested by referee #2.

(P3/L21-22 in BGD, P4/L4-6 in the revised MS) We have revised this sentence, as suggested by referee #2. Besides, we added a sentence at the end of this paragraph (P4/L11-13 in the revised MS) to present the hypothesis of this study more clearly, as suggested by referee #2.

(P4/L2 in BGD, P4/L15 in the revised MS) We have revised this sentence as suggested by referee #1.

(P4/L6 in BGD, P4/L19 in the revised MS) We have revised here as suggested by referee #1. Additionally, we added "most of" prior to "which is produced via photochemical reactions …", as suggested by referee #2.

(P4/L10 in BGD, P4/L24-25 in the revised MS) We added a sentence here to explain remineralized nitrate also applies to atmospheric nitrate that has been involved in the N cycle, as suggested by referee #1.

(P4/L12 in BGD, P5/L1-2 in the revised MS) As presented in our reply to the anonymous referee #1, we have revised here to use 0‰ for $\Delta^{17}O$ of $NO_3^-{}_{re}$, while citing the reference.

(P4/L15-17 in BGD, P5/L5-8 in the revised MS) As presented in our reply to the anonymous referee #1, what we wanted to say was that the geographical difference in the annual average $\Delta^{17}O$ values of $NO_3^-{}_{atm}$ was less than a few ‰ in mid-latitude. Thus, we have revised this sentence to clarify what we wanted to say.

(P4/L21 in BGD, P5/L12-13 in the revised MS) We have clarified what "partial metabolism" meant, in response to the comment from referee #1.

(P5/L7 in BGD, P5/L22-25 in the revised MS) We have revised here to clarify the reason we used the error in the $\Delta^{17}O$ value of $NO_3^-{}_{atm}$ (±3‰), in response to the comment from referee #1.

(P5/L11-12 in BGD, P5/L4-5 in the revised MS) We have reduced the number of citations here as suggested by referee #1.

(P5/L15 in BGD, P6/L8 in the revised MS) We have removed "continuous", as suggested by referee #2.

(P5/L16 in BGD, P6/L9 in the revised MS) We have specified the surface proportion of each watershed actually being covered by forests, in response to the referee #1's comment on P6/L1 of BGD.

(P5/L25 in BGD, P6/L20-23 in the revised MS) We have specified the respective contribution of precipitation in each season to the annual total here, in response to the referee #1's comment on P17/L21-23 in BGD.

(P6/L2 in BGD, P6/L26-P7/L1 in the revised MS) We have provided the loading rate of atmospheric N, as suggested by referee #1.

(P7/L7-9 in BGD, P8/L15-20 in the revised MS) We have provided the details of our sampling, as suggested by referee #1.

(P7/L13 in BGD, P8/L24 in the revised MS) We have fixed a typo here, as suggested by referee #1.

(P7/L19-22 in BGD, P9/L4 and L8 in the revised MS) We have provided the number of samples collected during the winter period vs the rest of the year, as suggested by referee #1.

(P7/L22 in BGD, P9/L8 in the revised MS) We have replaced "to" by "and", suggested by referee #2.

(P8/L14 in BGD, P10/L1-2 in the revised MS) We have added a sentence to notice that the error in the deposition rate in site KJ will be discussed in section 3.1, in response to the referee #1's comment to here and P14/L17 in BGD.

(P9/L1 in BGD, P10/L14-16 in the revised MS) We added a sentence here to present the reasons why the number of samples for stable isotopes were limited to about 1/2 of the whole at site KJ, in response to the referee #1's comment on Figure 3.

(P9/L15 in BGD, P11/L5 in the revised MS) We have removed ", the procedure [. . .]. Approximately", as suggested by referee #2.

(P10/L26 in BGD, P12/L15-16 in the revised MS) We have provided the highest uncertainty caused by presence of nitrite in a sample on the $\Delta^{17}O$ value of nitrate, as suggested by referee #1.

(P11/L14-P12/L13 (section 2.7) in BGD, P13/L5-P14/L5 in the revised MS) We added the numbering of the equations, which had been removed when we were arranging the format for Biogeosciences Discuss., in response to the referees' comments on P19/L16, etc.

(P12/L1 in BGD, P13/L17 in the revised MS) We have replaced "obtain" by "estimate", as suggested by referee #2.

(P12/L13 in BGD, P14/L5 in the revised MS) We have added "annual" before "deposition", as suggested by referee #1.

(P12/L16-21 in BGD) We have removed this paragraph as suggested by referee #2.

(P12/L25 and P14/L22 in BGD, P14/L19-21 and P17/L5-8 in the revised MS) We have added a sentence here to describe the variation ranges of $F_{total}$ and $F_{atm}$, as suggested by referee #2.

(P13/L6-7 in BGD, P15/L7-10 in the revised MS) We added a sentence and a reference here to compare our stream nitrate concentration in site KJ with those measured in the other forested catchments, as suggested by referee #1.

Shibata, H., Kuraji, K., Toda, H., and Sasa, K.: Regional Comparison of Nitrogen Export to

Japanese Forest Streams, The Scientific World Journal, 1, 572-580, doi:10.1100/tsw.2001.371, 2001.

(P13/L15 in BGD, P15/L18 in the revised MS) We removed "or more", as suggested by referee #1.

(P13/L16 in BGD, P15/L18-19 in the revised MS) In response to the referee #2's comment on section 2.3 (soil water sampling), we emphasized here in the revised MS that we could not find any significant differences in the isotopes of soil nitrate irrespective of the sampling depths and stations (Fig. 3) and thus we concluded that the data on isotopes obtained at SLS and SMS represented that of soil nitrate at the site KJ.

(P14/ L4-18 in BGD, P16/L8-22 in the revised MS) We added the values in the unit kg-N ha$^{-1}$ together with the values in the unit we used (mmol m$^{-2}$ yr$^{-1}$), in response to the referee #1's request.

(P14/L20-22 in BGD) We have removed this paragraph, as suggested by referee #2.

(P14/L22-26 in BGD, P15/L8-12 in the revised MS) We fixed typo found in the values in this paragraph.

(P15/L23 in BGD, P18/L11-14 in the revised MS) We added a sentence here to compare our stream nitrate concentration in sites IJ1 and IJ2 with those measured in the other forested catchments, as suggested by referee #1.

(P15/L23-26 in BGD, P18/L14-18 in the revised MS) We revised this sentence to avoid misleading readers, in response to the referee #1's comment on this sentence.

(P16/L17 in BGD, P19/L8 in the revised MS) We have specified the soil water sampled in SLS and SMS belongs to through flow, as suggested by referee #2.

(P16/L17-20 in BGD, P19/L9-12 in the revised MS) We revised this sentence, in response to the referee #1's comment.

(P17/L1 in BGD, P20/L3-4 in the revised MS) We revised this sentence in response to referee #2's comment.

(P17/L3-4 and Figure 4 in BGD, P20/L6-8 and Figure 4 in the revised MS) We revised here to emphasize that the regression line used to estimate the endmember $\delta^{18}$O value of $NO_3^-{}_{re}$ was obtained from the data of both stream nitrate and soil nitrate, in response to the referee #1's comment. We presented both the correlation coefficient ($r^2$) and the p-value of the regression line in text as well (P17/L3 in BGD). Besides, we added the mixing lines between the end-members ($NO_3^-{}_{atm}$ and $NO_3^-{}_{re}$) in the figure. As for the mixing line, not only the mixing line between $NO_3^-{}_{atm}$ and $NO_3^-{}_{re}$ having the average $\delta^{18}$O value but also that between $NO_3^-{}_{atm}$ and $NO_3^-{}_{re}$ having the lowermost $\delta^{18}$O value were presented so that we clarified this in the caption, as suggested by the referees. On the other hand, we removed the regression line used to estimate the endmember $\delta^{18}$O value of $NO_3^-{}_{re}$ from Fig. 4 as it is confusing to draw 3 lines in the figure.

(P17/L13 in BGD, P20/L15-24 in the revised MS) We presented how to estimate $\delta^{18}$O value of $NO_3^-{}_{re}$ together with the equation we used, in response to the request from referee #1.

(P17/L14-19 in BGD, P20/L24-P21/L8 and Figure 4 in the revised MS) We thoroughly revised here to emphasize that the oxygen isotopic fractionation through partial metabolism was minor in $NO_{3\ re}^{-}$ at KJ, in response to the referee #1's comment.

(P18/L6 in BGD, P21/L22-23 in the revised MS) We added references after the value of 0‰, as suggested by referee #1.

(P18/L18-19 in BGD, P22/L9-10 in the revised MS) We have revised this sentence to "stream nitrate concentration shows a normal correlation with soil nitrate concentration", as suggested by referee #1.

(P18/L22 in BGD, P22/L12-13 in the revised MS) We revised this sentence to "This delay time reflects the magnitude and flow of the nitrate. . .", as suggested by referee #2.

(P19/L5-12 in BGD, P22/L23-P23/L4 in the revised MS) We have fixed the typos in the values of $M_{total}$ and $M_{atm}$.

(P20/L7-8 and Figure 9 in BGD, P24/L6-9 and Figure 9 in the revised MS) We have added two sentences here to explain (1) that the arrows related to the $M_{atm}/D_{atm}$ ratios are shown in red/pink in Fig. 9, while those related to nitrification are shown in brown/yellow, and (2) the colour differences represented the $M_{atm}/D_{atm}$ ratios were determined independent of nitrification, in response to the referees' comment on P20/L7-8 in BGD. Additionally, we have revised Figure 9, together with the caption of this figure.

(P20/L17-22 in BGD, P25/L3-9 in the revised MS) We revised this paragraph to emphasize that the "artificial processes" we presented here correspond to the direct contamination processes of nitrate and thus the increases in stream nitrate concentrations due to secondary changes in the N cycles within forested catchments were not included, as suggested by referee #2.

(P21/L24 in BGD, P26/L11 in the revised MS) We revised this sentence to "… by 6 and 20 times respectively in accordance …", as suggested by referee #2.

(Figures 1 and 2) We used similar formats for scale bars etc. in the figures, in response to the referee #2's request.

(Figure 2) We removed the distances ("120 m" and "40 m") shown in the figure, in response to the referee #2's question.

(Figure 3) We added a new figure 3(e) in which $F_{atm}$ and $F_{total}$ for IJ2 are presented, in response to the comment from referee #1 on P14/L21-22 in BGD.

(References) We added Costa et al. (2011) in the reference list, which had been cited in the text but accidentally removed from the list.

Costa, A. W., Michalski, G., Schauer, A. J., Alexander, B., Steig, E. J., and Shepson, P. B.: Analysis of atmospheric inputs of nitrate to a temperate forest ecosystem from $\Delta^{17}O$ isotope ratio measurements, Geophys. Res. Lett., 38, doi:10.1029/2011GL047539, 2011.

(References) We removed Morin et al. (2009) from the reference list.

Morin, S., Savarino, J., Frey, M. M., Domine, F., Jacobi, H. W., Kaleschke, L., and Martins, J.

M. F.: Comprehensive isotopic composition of atmospheric nitrate in the Atlantic Ocean boundary layer from 65° S to 79° N, J. Geophys. Res., 114, doi:10.1029/2008jd010696, 2009.

We would like to thank you and referees for the helpful comments and suggestions. We trust that the revision is satisfactory response to the referees' comments. Thank you for your consideration.

Sincerely yours,
Urumu Tsunogai, PhD
Professor
Graduate School of Environmental Studies,
Nagoya University
Furo-cho, Chikusa-ku, Nagoya,
464-8601, JAPAN
Phone: +81-11-789-3498
E-mail: urumu@nagoya-u.jp

Encl.
c.c. Drs. Nakagawa, F., Obata, Y., Ando, K., Yamashita, N., Saito, T., Uchiyama, S., Morohashi, M., Sase, H.,

[revised manuscript text omitted]

**KJ site**

| ページ 44: [7] 削除 | Urumu Tsunogai | 2018/10/06 15:05:00 |
|---|---|---|

**KJ site**

---

## Referee Report (RR1)

**Overall comment:**

This is the revised version of the manuscript by Nakagawa et al. entitled "Export flux of unprocessed atmospheric nitrate from temperate forested catchments: a possible new index for nitrogen saturation". I would like to acknowledge the considerable effort undertaken by the authors to address the various issues I had raised in my previous evaluation. I feel that the revised manuscript is much stronger, more well written, and above all clearer than the first version. The authors did respond to my questions in a way that satisfied me, and I now understand what they were trying to do in this work. Here, they use the Matm/Datm ratio as an independent way to assess the nitrogen saturation status of forested catchments in Japan. They applied this method on three catchments that have been heavily monitored in the past, and for which they had already a fair knowledge of the nitrogen saturation status. They find that the use of the Matm/Datm ratio provides a good indication of the relative saturation level between forested catchments. While I still have a few comments, I don't see why this manuscript shouldn't be published in Biogeosciences.

**Main comment:**

My main interrogation now is regarding your explanation of the difference in the Matm/Datm ratios between the studied catchments: you say that the difference is due to different biological assimilation rates, caused by different nitrogen saturation status. But couldn't the different biological assimilation rate be caused only by the difference in the vegetation in the forest for each catchment? The tree species abundances are different in KJ, IJ1 and IJ2 forests: could that cause the difference in the Matm/Datm ratios because the trees would have different metabolic rates? And in that case, the Matm/Datm ratio would not be an indicator of N saturation, but just of the retention capacity of a catchment? I think this needs to be at least discussed in your last section.

**Specific comments**

**Section 2.3:** Please expand a bit on your methods here. Explain for instance the conditioning of the samples (fridge, freezer, filtration) and if you use acid solution to stop any biological activity, etc.

**P.7 line 10 and P.8 line 11:** Please refer to Aber et al. (1989) when mentioning N saturation.

Aber, J. D., Nadelhoffer, K. J., Steudler, P. and Melillo, J. M.: Nitrogen saturation in northern forest ecosystems, BioScience, 39(6), 378–386, doi:10.2307/1311067, 1989.

**P.15 l.3:** Please refer to Bourgeois et al. (2018) when mentioning NO3-atm export from the snowpack to the river.

Bourgeois, I., Savarino, J., Caillon, N., Angot, H., Barbero, A., Delbart, F., Voisin, D. and Clément, J.-C.: Tracing the fate of atmospheric nitrate in a subalpine watershed using $\Delta^{17}O$, Environ. Sci. Technol., 52(10), 5561–5570, doi:10.1021/acs.est.7b02395, 2018.

**P.19 l.11:** Please remove "huge", or give an idea of the reservoir of nitrate it represents if you know it. No need to emphasize on the size, we understand the idea of dilution here.

**P.21 l.4:** Please replace "fractionation" with "enrichment"

**Table 1:** Maybe add a row with Matm/Datm ratio for each catchment

---

## Referee Report (RR2)

g-2018-258    Submitted on 31 May 2018 –Revised version

Export flux of unprocessed atmospheric nitrate from temperate forested catchments: A possible new index for nitrogen saturation

Fumiko Nakagawa, Urumu Tsunogai, Yusuke Obata, Kenta Ando, Naoyuki Yamashita, Tatsuyoshi Saito, Shigeki Uchiyama, Masayuki Morohashi, and Hiroyuki Sase

The authors thoroughly revised their manuscript following most of the comments. As result I think the paper is now acceptable for publication. Yet, in the process of reviewing the revised ms I found some new typos:

P5-L1: ‰ is missing.

P8-L19: twice (not thrice).

P14-L8: there is something wrong with this sentence, too many copy and paste I guess.

This means there will be minor revisions so I take this opportunity to ask again:

1- Please give the units for each Equation.
2- In the discussion please specify to which category (catchment groundwater, through flow) belongs the water sampled in SLS and SMS.
3- A figure showing the relationship between Matm and stream NO3 conc. would be welcome to support your statement P23-L20-21.

---

## Author Response (AR2)

Nov. 1, 2018

Dr. Sébastien Fontaine
Editor of Biogeosciences

Title: Export flux of unprocessed atmospheric nitrate from temperate forested catchments: A possible new index for nitrogen saturation
Authors: Nakagawa, F., Tsunogai, U., Obata, Y., Ando, K., Yamashita, N., Saito, T., Uchiyama, S., Morohashi, M., Sase, H.,
MS No.: bg-2018-251

Dear Dr. Fontaine:

Thank you very much for your e-mail on 1 Nov., 2018. We are very glad to hear that our manuscript was accepted with corrections. We have carefully studied the additional comments and corrected the manuscript. All the corrections have been listed in this letter. Besides, we have uploaded the revised manuscript.

Revisions from the previous submission to BG are as follows:

(P5/L1 in the previous/corrected MS) We added "‰", as suggested by referee #2.

(P7/L10 in the previous/corrected MS) We cited Aber et al. (1989) here, as suggested by referee #1.

(P8/L11 in the previous/corrected MS) We cited Aber et al. (1989) here, as suggested by referee #1.

(P8/L19 in the previous/corrected MS) We replaced "thrice" by "twice", as suggested by referee #2.

(P9/L4 in the previous/corrected MS) We added several words here to explain the details of the soil water sampling, as suggested by referee #1.

(P14/L25 and the reference list in the previous/corrected MS) We added a new reference here (Bourgeois et al., EST, 2018a), as suggested by referee #1. Due to the revision, we also replaced Bourgeois et al. (2018) in the previous MS (Bourgeois et al., Sci. Total Environ., 2018) by Bourgeois et al. (2018b).

Bourgeois, I., Savarino, J., Caillon, N., Angot, H., Barbero, A., Delbart, F., Voisin, D., and Clément, J.-C.: Tracing the fate of atmospheric nitrate in a subalpine watershed using $\Delta^{17}O$, Environ. Sci. Technol., 52, 5561−5570, 2018a.

(P19/L4 in the previous/corrected MS) We added several words here to emphasize that the "soil nitrate" corresponds to nitrate in the soil water samples (SLS20, SLS60, and SMS20), in response to the referee #2's comments.

(P19/L11 in the previous MS, P19/L12 the corrected MS) We removed "huge", as suggested by referee #1.

(P21/L4 in the previous MS, P21/L6 the corrected MS) We replaced "oxygen isotopic fractionation" by "$^{18}O$-enrichment", as suggested by referee #1.

(P24/L15 and P27/L12 in the previous MS, P24/L17 and P27/L15 in the corrected MS) We revised sections 3.5 and 4 to emphasize that, not only the $M_{atm}/D_{atm}$ ratio, but also the stream nitrate concentration is the index of the nitrogen saturation stages, in response to the referee #1's comments.

(Table 1 in the previous/corrected MS) We added a row for the $M_{atm}/D_{atm}$ ratios, in response to the referee #1's comments.

The answers to the referees' questions/comments are as follows:

**1> Section 2.3: Please expand a bit on your methods here. Explain for instance the**
**1> conditioning of the samples (fridge, freezer, filtration) and if you use acid solution to stop**
**1> any biological activity, etc.**

We have added several words in section 2.3 to explain the details on the soil water sampling. The conditioning of the soil water samples, however, had been explained in section 2.5 (Analysis) in the previous/corrected MS, so that we did not add explanation on the conditioning here.

**1> Main comment:**
**1> My main interrogation now is regarding your explanation of the difference in**
**1> the Matm/Datm ratios between the studied catchments: you say that the difference is due**
**1> to different biological assimilation rates, caused by different nitrogen saturation status.**
**1> But couldn't the different biological assimilation rate be caused only by the difference**
**1> in the vegetation in the forest for each catchment? The tree species abundances are different**
**1> in KJ, IJ1 and IJ2 forests: could that cause the difference in the Matm/Datm ratios because**
**1> the trees would have different metabolic rates? And in that case, the Matm/Datm ratio**
**1> would not be an indicator of N saturation, but just of the retention capacity of a catchment?**
**1> I think this needs to be at least discussed in your last section.**

Please note that, not only the $M_{atm}/D_{atm}$ ratios, but also the stream nitrate concentration is the index of the nitrogen saturation stages. If the differences in vegetation were responsible for the high $M_{atm}/D_{atm}$ ratios in the studied sites (KJ site, especially) as you insisted, the differences in vegetation should be responsible for the elevated stream nitrate concentrations in the studied sites (KJ site, especially) as well. While there were many past studies on the elevated stream nitrate concentrations in the studied sites, as well as the other forested watersheds in the world as already presented in the MS, none of them had found significant correlation between stream nitrate concentrations and the differences in vegetation. That is to say, while the differences in vegetation could explain the variations in $M_{atm}/D_{atm}$ ratios, it is impossible to explain the normal correlation between stream nitrate concentrations and $M_{atm}/D_{atm}$ ratios in Fig. 8(a).

We revised sections 3.5 and 4 to emphasize that, not only the $M_{atm}/D_{atm}$ ratio, but also the stream nitrate concentration is the index of the nitrogen saturation, in response to your comment.

**2> 1- Please give the units for each Equation.**

We already answered to this comment during the discussion stage. Because the equations in this manuscript were general equations and no actual value was presented in the equations, there are no constraints on the units. That is to say, we can use any unit we want.

If we used "µmol $L^{-1}$" for $C_{atm}$ and "L $day^{-1}$" for V and "$m^2$" for S, for instance, the unit of $F_{atm}$ became "µmol $m^{-2}$ $day^{-1}$" using equation (3). If we used "mg $L^{-1}$" for $C_{atm}$ and "L $yr^{-1}$" for V and "$km^2$" for S, for instance, the unit of $F_{atm}$ became "mg $km^{-2}$ $yr^{-1}$".

**2> 2- In the discussion please specify to which category (catchment groundwater, through**
**2> flow) belongs the water sampled in SLS and SMS.**

We already answered to this question during the discussion stage. We interpreted that the soil water sampled in SLS and SMS belongs to through flow.

In response to your question during the discussion stage, we already emphasized that soil nitrate represents nitrate in through flow (P19/L8) during the previous revisions. Additionally, we newly added several words in section 3.3 to emphasize that the soil nitrate corresponds to nitrate in the soil water samples (SLS20, SLS60, and SMS20).

**2> 3- A figure showing the relationship between Matm and stream NO3 conc. would be**
**2> welcome to support your statement P23-L20-21.**

In response to the same request from you during the discussion stage, we had added Fig. 8(c) during the previous revisions.

We would like to thank you and referees for the helpful comments and suggestions. We trust that the revision is satisfactory response to the referees' comments. Thank you for your consideration.

Sincerely yours,
Urumu Tsunogai, PhD
Professor
Graduate School of Environmental Studies,
Nagoya University
Furo-cho, Chikusa-ku, Nagoya,
464-8601, JAPAN
Phone: +81-11-789-3498
E-mail: urumu@nagoya-u.jp

Encl.
c.c. Drs. Nakagawa, F., Obata, Y., Ando, K., Yamashita, N., Saito, T., Uchiyama, S., Morohashi, M., Sase, H.,